# NIERT: Accurate Numerical Interpolation through Unifying Scattered Data Representations using Transformer Encoder

## Abstract

Numerical interpolation for scattered data, i.e., estimating values for target points based on those of some observed points, is widely used in computational science and engineering. The existing approaches either require explicitly pre-defined basis functions, which makes them inflexible and limits their performance in practical scenarios, or train neural networks as interpolators, which still have limited interpolation accuracy as they treat observed and target points separately and cannot effectively exploit the correlations among data points. Here, we present a learning-based approach to numerical interpolation for scattered data using encoder representation of Transformers (called NIERT). Unlike the recent learning-based approaches, NIERT treats observed and target points in a unified fashion through embedding them into the same representation space, thus gaining the advantage of effectively exploiting the correlations among them. The specially-designed partial self-attention mechanism used by NIERT makes it escape from the unexpected interference of target points on observed points. We further show that the partial self-attention is essentially a learnable interpolation module combining multiple neural basis functions, which provides interpretability of NIERT. Through pre-training on large-scale synthetic datasets, NIERT achieves considerable improvement in interpolation accuracy for practical tasks. On both synthetic and real-world datasets, NIERT outperforms the existing approaches, e.g., on the TFRD-ADlet dataset for temperature field reconstruction, NIERT achieves an MAE of $1.897 \times 10^{-3}$, substantially better than the state-of-the-art approach (MAE: $27.074 \times 10^{-3}$). The source code of NIERT is available at https://anonymous.4open.science/r/NIERT-2BCF.

## 1 Introduction

Scattered data consist of a collection of points and corresponding values, in which the points have no structure besides their relative positions (Franke & Nielson, 1991). Scattered data arise naturally and widely from a large variety of theoretical and practical scenarios, including solving partial differential equations (PDEs) (Franke & Nielson, 1991; Liu, 2016), temperature field reconstruction (Chen et al., 2021), and time series interpolation (Lepot et al., 2017; Shukla & Marlin, 2019). These scenarios usually require numerical interpolation for scattered data, i.e., estimating values for target points based on those of some observed points. For example, in the task of temperature field reconstruction for micro-scale electronics, interpolation methods are used to obtain the real-time working environment of electronic components from limited measurements, and imprecise interpolation might significantly increase the cost of predictive maintenance. Thus, accurate approaches to numerical interpolation are highly desirable.

A large number of approaches have been proposed for interpolating scattered data. Traditional approaches use schemes that approximate the target function by a linear combination of some basis functions (Heath, 2018), in which the basis functions should be explicitly pre-defined. To adapt to different scenarios, various types of basis functions have been devised. These schemes can theoretically guarantee the interpolation accuracy when sufficient observed points are available; however, they have also been shown to be ineffective for sparse data points (Bulirsch et al., 2002). In addition, the schemes can hardly learn from the experience of interpolation in similar tasks.

Recent progress has exhibited an alternative strategy that uses neural networks to learn target functions directly from the given observed points. For example, conditional neural processes (CNPs) (Garnelo et al., 2018) and their extensions (Kim et al., 2019; Lee et al., 2020b;a) model the conditional distribution of regression functions given the observed points, and Chen et al. (2021) proposed to use vanilla Transformer (Vaswani et al., 2017) to solve interpolation task in temperature field reconstruction. All of these approaches use an "encoder-decoder" architecture, in which the encoder learns the representations of observed points while the decoder estimates values for target points. Ideally, observed points and target points should be processed in a unified fashion because they are from the same domain. However, these approaches treat them separately and cannot effectively exploit the correlation between them.

Here, we present an approach to numerical interpolation that can effectively exploit the correlations between observed points and target points. Our approach is a learning-based approach using the encoder representations of Transformers (thus called NIERT). The key elements of NIERT include: $i$) the use of mask mechanism, which complements target points with learnable mask tokens as performed by masked language model, say BERT (Devlin et al., 2018), and thus enables processing both observed and target points in a unified fashion, $ii$) a novel partial self-attention model that calculates attentions among the given data points except for the influence of target points onto the observed points at each layer, thus gaining the advantages of exploiting the correlations between these two types of points and, more importantly, avoiding the unexpected interference of target points on observed points at the same time, and $iii$) the use of the pre-training technique, which leverages large-scale low-cost synthetic data to build powerful, general-purpose, and transferable pre-trained interpolation models. The use of the pre-trained models significantly improves generalization ability and interpolation accuracy.

The main contributions of this study are summarized as follows.

- We propose an accurate approach to numerical interpolation for scattered data. On representative datasets including synthetic and real-world datasets, our approach outperforms the state-of-the-art approaches and shows potential in a wide range of application fields.
- We propose a novel partial self-attention mechanism to make Transformer incorporated with strong inductive bias for interpolation tasks, i.e., it can effectively exploit the correlation among two types of points and, at the same time, avoids the interference of one type of points onto the others.
- We also demonstrate the essence of NIERT, i.e., a learnable interpolation approach using neural basis functions, through illustrating the deep connection between partial self-attention mechanism and traditional interpolation approaches.
- To the best of our knowledge, this study is the first work to propose the pre-trained models for scatter-data interpolation. We have verified that such interpolation pre-trained models can be generalized to a wide range of interpolation tasks.

## 2 RELATED WORKS

### 2.1 TRADITIONAL INTERPOLATION APPROACHES FOR SCATTERED DATA

Traditional interpolation approaches for scattered data use explicitly pre-defined basis functions to construct interpolation function, e.g., Lagrange interpolation, Newton interpolation (Heath, 2018), B-spline interpolation (Hall & Meyer, 1976), Shepard's method (Gordon & Wixom, 1978), Kriging (Wackernagel, 2003), and radial basis function interpolation (RBF) (Powell, 1987; Fornberg & Zuev, 2007). Among these approaches, the classical Lagrange interpolation, Newton interpolation and B-splines interpolation are usually used for univariate interpolation. Wang et al. (2010) proposed a high-order multivariate approximation scheme for scattered data sets, in which approximation error is represented using Taylor expansions at data points, and basis functions are determined through minimizing the approximation error.

### 2.2 NEURAL NETWORK-BASED INTERPOLATION APPROACHES

Equipped with deep neural networks, data-driven interpolation and reconstruction methods show great advantages and potential. For instance, convolutional neural networks (CNNs) have been ap-

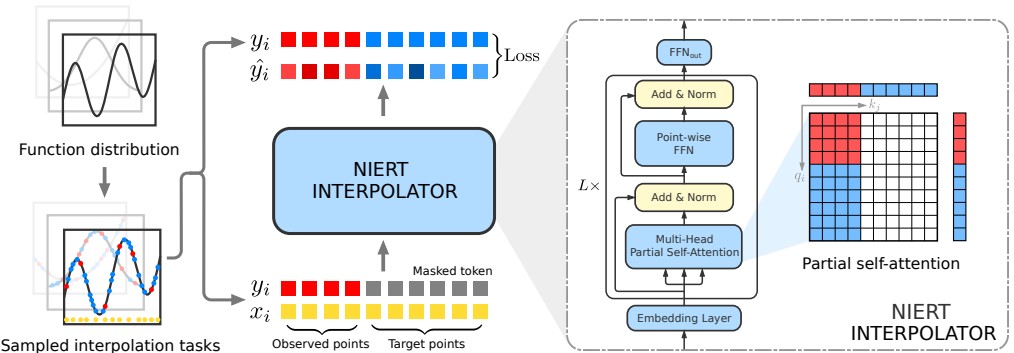

Figure 1: Overview of NIERT training process. Here, $x_i$ represents the position of a point, and $y_i$ represents its value. The predicted values of the point is denoted as $\hat{y}_i$. We prepare the training interpolation tasks by first sampling functions from a function distribution $\mathcal{F}$ and then sampling observed points $O$ and target points $T$ on each function. NIERT trains an interpolator over this data. The partial self-attention mechanism facilitates exploiting the correlations between observed points and target points and avoiding the unexpected interference of target points on observed points

plied in the interpolation tasks of single image super-resolution (Tai et al., 2017; Li et al., 2021), and recurrent neural networks (RNNs) and Transformers have been used for interpolation of sequences like time series data (Shukla & Marlin, 2019; 2021).

Recently, Garnelo et al. (2018) proposed to model the conditional distribution of regression functions given observed points. The proposed approach, called conditional neural processes (CNPs), has shown increased estimation accuracy and generalizing ability. A variety of studies have been performed to enhance CNPs: Kim et al. (2019) designed attentive neural processes (ANPs), which shows improved accuracy. Lee et al. (2020a) leveraged Bayesian last layer (BLL) (Weber et al., 2018) for faster training and better prediction. In addition, the bootstrap technique was also employed for further improvement (Lee et al., 2020b). To solve the interpolation task in 2D temperature field reconstruction, Chen et al. (2021) proposed a Transformer-based approach, referred to as TFR-transformer, which can also be applied to solve interpolation tasks for scattered data with higher dimensions.

## 2.3 MASKED LANGUAGE/IMAGE MODELS AND THE PRE-TRAINING TECHNIQUE

The design of NIERT is also inspired by the recent advances in masked language/image models (Devlin et al., 2018; Bao et al., 2021; Brown et al., 2020; He et al., 2021) and pre-trained models for symbolic regression (Biggio et al., 2021; Valipour et al., 2021). The mask mechanism enables the models to reconstruct missing data from their context and effectively learn representations of language and images. In addition, the pre-trained models for symbolic regression were designed to learn the map from scattered data to corresponding symbolic formulas (Biggio et al., 2021; Valipour et al., 2021). Here, NIERT uses the mask mechanism to reconstruct the missing values of target points and uses the pre-training technique to learn the experience of interpolation on similar tasks.

## 3 METHOD

### 3.1 OVERVIEW OF NIERT

In the study, we focus on the interpolation task that can be formally described as follows: We are given $n$ observed points with known values $O = \{(x_i, y_i)\}_{i=1}^n$, and $m$ target points with values to be determined, denoted as $T = \{x_i\}_{i=n+1}^{n+m}$. Here, $x_i \in X$ denotes position of a point, $y_i = f(x_i) \in Y$ denotes the value of a point, and $f : X \to Y$ denotes a function mapping positions to values. The function $f$ is from a function distribution $\mathcal{F}$, which can be explicitly defined using a mathematical formula or implicitly represented using a set of scattered data in the form $(x_i, y_i)$. The goal of the

interpolation task is to accurately estimate the values $f(x)$ for each target point $x \in T$ according to the observed points in $O$.

The main element of our NIERT approach is a neural interpolator that learns to estimate values for target points. We train the neural interpolator using a set of interpolation tasks sampled from the function distribution $\mathcal{F}$ (Fig. 1). For each of the sampled interpolation tasks, we mask the values of target points, feed the task to the neural interpolator, and train the interpolator to predict the values of target points and observed points as well. The training objective is to minimize the error between the predicted values and the ground-truth values of both target and observed points. The trained interpolator can be used to interpolate values of target points for the interpolation tasks from the same function distribution.

## 3.2 ARCHITECTURE OF THE NIERT INTERPOLATOR

The neural interpolator in NIERT adopts the Transformer encoder framework; however, to suit the interpolation task, significant modifications and extensions were made in the embedding, Transformer and output layers, which are described in detail below.

**Embedding with masked tokens:** NIERT embeds both observed points and target points into the unified high-dimensional embedding space. As the position $x$ of a data point and its value $y$ are from different domains, we use two linear modules : $\text{Linear}_x$ embeds the positions while $\text{Linear}_y$ embeds the values.

It should be noted that for target points, their values are absent when embedding as they are to be determined. In this case, we use a masked token as substitute, which is embedded as a trainable parameter $\text{MASK}_y$ as performed in BERT (Devlin et al., 2018). This way, the interpolator processes both target points and observed points in a unifying fashion.

We concatenate the embeddings of position and value of a data point as the point's embedding, denoted as $h_i^0$, i.e.,

$$h_i^0 = \begin{cases} [\text{Linear}_x(x_i), \text{Linear}_y(y_i)], & \text{if } (x_i, y_i) \in O \\ [\text{Linear}_x(x_i), \text{MASK}_y], & \text{if } x_i \in T \end{cases}.$$

**Transformer layer with partial self-attention mechanism:** NIERT feeds the embeddings of the points into a stack of $L$ Transformer layers, producing encodings of these points as results. Each Transformer layer contains two subsequent sub-layers, namely, a multi-head self-attention module, and a point-wise fully-connected network. These sub-layers are interlaced with residual connections and layer normalization between them.

To avoid the unexpected interference of target points on observed points and target points themselves, NIERT replaces the original self-attention in Transformer layer with a *partial self-attention*, which calculates the feature of point $i$ at the $l + 1$-st layer as follows:

$$h_i^{l+1} = \text{LayerNorm}(\tilde{v}_i^l + \text{MLP}(\tilde{v}_i^l)), \tag{1}$$

$$\text{where } \tilde{v}_i^l = \text{LayerNorm}\left(v_i^l + \sum_j w_{ij}\alpha_{ij}^l v_j^l\right). \tag{2}$$

Here, $v_i^l$ and $\alpha_{ij}^l$ represent the value vector and ordinary attention weights at the $l$-th layer as calculated in Transformer (Vaswani et al., 2017). We use an additional term $w_{ij}$ to represent which entries of attention weights are neglected in the partial self-attention mechanism, i.e.,

$$w_{ij} = \begin{cases} 1, & \text{if } (x_j, y_j) \in O \\ 0, & \text{if } x_j \in T \end{cases}.$$

For each observed point $(x_j, y_j) \in O$, we set $w_{ij} = 1$; thus, NIERT can model the correlation between observed points and target points, and the correlation among observed points as well. Relatively, the correlation among observed points is easier to learn as these points have known values. In addition, this correlation can be transferred onto the target points, thus promoting learning representations of these data points.

In contrast, by forcing the weight $w_{ij}$ to be 0 for a target point $i$ and any point $j$, we completely avoid the unexpected interference of target points on the other points.

**Estimating values for target points:** For each target point $i$, we estimate its value $\hat{y}_i$ through feeding its features at the final Transformer layer into a fully connected feed-forward network, i.e.,

$$\hat{y}_i = \text{MLP}_{\text{out}}(h_i^L).$$

We calculate the error between the estimation and the corresponding ground-truth value, and compose the errors for all points into a loss function to be minimized.

### 3.3 ENHANCING NIERT USING PRE-TRAINING TECHNIQUE

The existing data-driven interpolation approaches, although powerful, still suffer from the shortcomings of limited generality and transferability. To overcome these shortcomings, we propose to use the pre-training technique to build NIERT into pre-trained interpolation models. Due to the lack of large-scale pre-training datasets for interpolation, we constructed the NeSymReS dataset , which contains a large variety of synthetic symbolic mathematical functions (see B.1 for further details). These functions approximate general and diverse function distribution, thus gaining our pre-trained model strong generalization ability. Here, we pre-train NIERT using NeSymReS dataset and fine-tune it on other datasets in practical scenarios.

### 3.4 DEEP CONNECTION BETWEEN NIERT AND CLASSICAL INTERPOLATION METHOD

The underlying rational of using partial self-attention mechanism is rooted in its deep connection with classical interpolation methods. We use RBF as a representative of classical methods to illustrate this connection below.

To interpolate $n$ observed points $O = \{(x_i, y_i)\}_{i=1}^n$, the RBF approach uses the following interpolation function

$$f_{\text{RBF}}(x) = \sum_{j=1}^n \lambda_j \phi(x, x_j). \tag{3}$$

Here, $\phi(\cdot, x_j)$ represents a radial basis function specified by the observed point $x_j$, and $\lambda_j$ is the coefficient, which can be determined through solving the linear equations $\sum_{j=1}^n \lambda_j \phi(x_i, x_j) = y_i$, $(x_i, y_i) \in O$ (Solomon, 2015).

We further rewrite the partial self-attention shown in Eq. (2) with simplification but without loss of its essence as:

$$f_{\text{Attn}}(x) = \sum_{j=1}^n \alpha(q(x), k_j) v_j. \tag{4}$$

Here, $\alpha(q(x), k_j)$ is the normalized attention weight that models the contribution to query vector $q(x)$ by the key vector $k_j$, in which $k_j$ and $v_j$ are dominated by the observed point $x_j$.

Comparing Eq. (3) with Eq. (4), we can find that partial self-attention is a general form of RBF interpolation function: $\lambda_j$ in Eq. (3) corresponds to $v_j$ in Eq. (4), and $\phi(x, x_j)$ in Eq. (3) corresponds to $\alpha(q(x), k_j)$ in Eq. (4). This correspondence indicates that $\alpha(q(x), k_j)$ can be treated as a learnable basis function and, similarly, $v_j$ can be treated as a predictable basis function coefficient. From this point of view, partial self-attention is a learnable layer that interpolates the representation of observed points to yield new representations of data points. Together, this insight provides a plausible explanation of the partial self-attention mechanism, which is also supported by the contribution analysis of observed points shown in Section 4.4.

It should be pointed out that the decoder of TFR-transformer can also be rewritten into a formula similar to Eq. (4); however, it still differs greatly from the partial self-attention mechanism: partial self-attention encodes both observed and target points into the same feature space at each layer whereas TFR-transformer treats these two types of data points separately. This unifying representation of data points gains NIERT the advantages of exploiting the correlations among data points more effectively.

## 4 EXPERIMENTS AND RESULTS

We evaluated NIERT and compared it with ten representative scattered data interpolation approaches on both synthetic and real-world datasets. We further examined the effects of the key elements of

NIERT, including the partial self-attention, and the pre-training technique. We also visualized the attention map to exhibit the contribution of each observed point to the interpolation function.

## 4.1 EXPERIMENT SETTING

We evaluated NIERT on three synthetic datasets (NeSymReS, TFRD-ADlet, and D30) and one real-world dataset (PhysioNet): NeSymReS is a synthetic dataset of up to 4 dimensions for mathematical function interpolation, TFRD-ADlet is a synthetic dataset for 2D temperature field reconstruction, and D30 is a high dimension dataset that contains data points of up to 30 dimensions. PhysioNet is a real-world dataset collected from intensive care unit (ICU) records for time-series data interpolation.

In this study, the NeSymReS dataset was used for pre-training NIERT to further improve its interpolation accuracy on TFRD-ADlet and PhysioNet datasets. For the TFRD-ADlet dataset, we directly use the 2D NeSymReS dataset for pre-training. As the PhysioNet dataset has a dependent variable with a dimensionality of 37, we construct the pre-training instances by stacking random 37 functions from the 1D NeSymReS dataset and then sampling interpolation task instances.

When evaluating NIERT and other interpolation approaches, the prediction error of target points is calculated as interpolation accuracy. For the NeSymReS, D30 and PhysioNet datasets, we adopted mean squared error (MSE) as the error metric. For the TFRD-ADlet dataset, we use three error metrics: mean absolute error (MAE), MAE in the component area (CMAE) and MAE at the boundary (BMAE) following Chen et al. (2021). Accordingly, we use $L_2$-form loss function for NeSymReS and PhysioNet dataset and $L_1$-form loss function for TFRD-ADlet dataset for training.

We compared NIERT with state-of-the-art approaches. Further details of datasets, approaches for comparison, and other experiment settings are provided in the Supplementary material.

## 4.2 INTERPOLATION ACCURACY ON SYNTHETIC AND REAL-WORLD DATASETS

For each instance of test dataset, we applied the trained NIERT to estimate values for target points. We calculate the errors between estimation and ground-truth as interpolation accuracy.

| Interpolation approach | MSE ($\times 10^{-5}$) on NeSymReS test set | | | |
|---|---|---|---|---|
| | 1D | 2D | 3D | 4D |
| RBF | 215.439 | 347.060 | 443.094 | 327.775 |
| MIR | 67.281 | 274.601 | 448.933 | 342.997 |
| CNP | 67.176 | 248.668 | 392.348 | 314.311 |
| ANP | 34.558 | 140.005 | 206.699 | 164.751 |
| BANP | 14.913 | 84.187 | 143.518 | 140.288 |
| TFR-transformer | 15.556 | 58.569 | 99.986 | 90.579 |
| NIERT | **8.964** | **45.319** | **77.664** | **72.025** |

Table 1: Interpolation accuracy of NIERT and existing approaches on NeSymReS dataset

| Interpolation approach | Evaluation criteria ($\times 10^{-3}$) | | |
|---|---|---|---|
| | MAE | CMAE | BMAE |
| CNP | 96.674 | 109.419 | 56.939 |
| ANP | 54.684 | 62.511 | 26.524 |
| BANP | 28.671 | 29.450 | 19.984 |
| TFR-transformer | 27.074 | 29.772 | 18.835 |
| NIERT | 3.473 | 3.947 | 2.467 |
| NIERT w/ pretraining | **1.897** | **1.971** | **1.246** |

Table 2: Interpolation accuracy of NIERT and existing approaches over TFRD-ADlet dataset

| Interpolation approach | MSE ($\times 10^{-4}$) on D30 test set | | |
|---|---|---|---|
| | 10D | 20D | 30D |
| RBF | 181.744 | 161.261 | 112.902 |
| MIR | 161.474 | 155.038 | 92.278 |
| CNP | 35.623 | 51.166 | 58.312 |
| ANP | 12.578 | 11.401 | 17.545 |
| BANP | 12.077 | 13.010 | 17.462 |
| TFR-transformer | 7.465 | 6.211 | 6.334 |
| NIERT | **5.496** | **4.681** | **1.549** |

Table 3: Interpolation accuracy of NIERT and existing approaches on D30 dataset

| Interpolation approach | Ratio of observed points | | |
|---|---|---|---|
| | 50% | 70% | 90% |
| RNN-VAE | 13.418±0.008 | 11.887±0.005 | 11.470±0.006 |
| L-ODE-RNN | 8.132±0.020 | 8.171±0.030 | 8.402±0.022 |
| L-ODE-ODE | 6.721±0.109 | 6.798±0.143 | 7.142±0.066 |
| mTAND-Full | 4.139±0.029 | 4.157±0.053 | 4.798±0.036 |
| NIERT | 2.868±0.021 | 2.656±0.041 | 2.709±0.157 |
| NIERT w/ pretraining | **2.831±0.021** | **2.641±0.052** | **2.596±0.159** |

Table 4: Relationship between interpolation accuracy (MSE, $\times 10^{-3}$) on PhysioNet with observed ratio

**Accuracy on NeSymReS and D30 datasets:** As shown in Table 1, on the 1D NeSymReS test set, RBF shows the largest interpolation error (MSE: 215.439). MIR, another approach using explicit basis functions, also shows a high interpolation error of 67.281. In contrast, BANP and TFR-transformer, which use neural networks to learn interpolation, show relatively lower errors (MSE: 14.913, 15.556). Compared with these approaches, our NIERT approach achieves the best interpolation accuracy (MSE: 8.964). Table 1 also demonstrates the advantage of NIERT over the existing approach on the 2D, 3D, and 4D instances.

To examine in depth the interpolation accuracy, we further divide test instances into subsets according to the number of observed points. As shown in Figure 2, as the number of observed points increases, the interpolation error decreases as expected. In addition, the relative advantages of these approaches vary with the number of observed points, e.g., CNP is better than RBF and MIR initially but finally becomes worse as the number of observed points increases. Among all approaches, NIERT stably shows the best performance over all test subsets, regardless of the number of observed points.

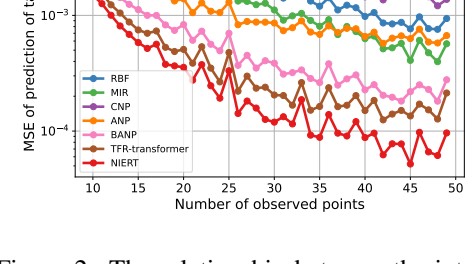

Figure 2: The relationship between the interpolation accuracy and the number of observed points. Here we use the 2D instances in the NeSymReS test dataset as representatives

Table 3 suggests that even for D30 dataset with high-dimensional (up to 30 dimensions) data points, NIERT still shows excellent interpolation accuracy and outperforms all existing methods, which demonstrates the scalability of NIERT to the high-dimensional interpolation tasks.

**Accuracy on TFRD-ADlet and PhysioNet datasets:** As shown in Table 2, CNP, although employing the neural network technique, still performs poorly with MAE as high as 96.674. In contrast, NIERT achieves the lowest interpolation error (MAE: 3.473), which is over one order of magnitude lower than CNP, ANP, BANP and TFR-transformer. Moreover, when enhanced with the pre-training technique, NIERT can further decrease its interpolation MAE to 1.897. Besides MAE, other metrics, say CMAE and BMAE, also show the superior of NIERT over the existing approaches (Table 2).

Table 4 suggests that on the PhysioNet dataset, NIERT also outperforms the existing approaches, e.g., when controlling the ratio of observed points to be 50%, NIERT achieves an average MSE ($\times 10^{-3}$) of 2.868, significantly lower than other approaches. Again, NIERT with the pre-training technique shows better performance. The advantages of NIERT hold across various settings of the ratio of the observed points.

Taken together, these results demonstrate the power of NIERT for numerical interpolation in multiple application fields, including interpolating the scattered data generated using mathematical functions, reconstructing temperature fields, and interpolating time-series data.

## 4.3 Case studies of interpolation results

To further understand the advantages of NIERT, we carried out case studies through visualizing the observed points, the reconstructed interpolation functions and the interpolation errors in this subsection. More visualized cases are put in Section D.1 in the Supplementary material.

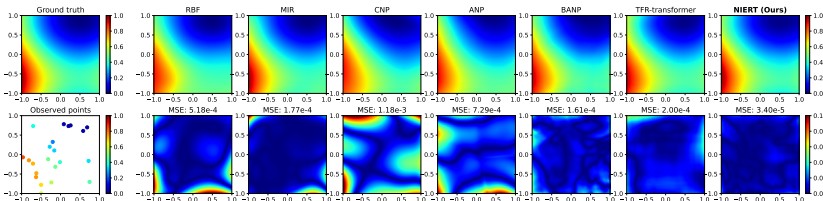

Figure 3: An example of 2D interpolation task extracted from NeSymReS test set. The up-left figure shows the ground-truth function while the bottom-left figure shows the 22 observed points. The interpolation functions reported by NIERT and the existing approaches are listed on the top panel with their differences with the ground-truth are list below

Figure 3 show a 2D instance in the NeSymReS test set, respectively. As illustrated, RBF performs poorly in the application scenario with sparse observed data. In addition, RBF and MIR, especially ANP, cannot accurately predict values for the target points that fall out of the range restricted by observed points. The CNP approach can only learn the rough trend stated by the observed points, thus leading to significant errors. In contrast, BANP, TFR-transformer and NIERT can accurately

estimate values for target points within a considerably large range, and compared with BANP and TFR-transformer, NIERT can produce more accurate results.

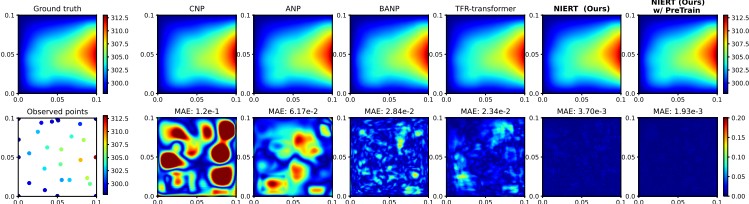

Figure 4: An example of temperature field reconstruction task extracted from TFRD-ADlet test set. The up-left figure shows the ground-truth temperature field while the bottom-left figure shows the 32 observed points. The reconstructed results reported by NIERT and the existing approaches are listed on the top panel with their differences with the ground-truth temperature field are list below

Figure 4 shows an instance of temperature field reconstruction extracted from TFRD-ADlet. From this figure, we can observe that when using the pre-training technique, NIERT further improves its interpolation accuracy in the whole area.

## 4.4 CONTRIBUTION ANALYSIS OF OBSERVED POINTS FOR INTERPOLATION

An idealized interpolation approach is expected to effectively exploit all observed points with appropriate consideration of relative positions among observed points and target points as well. To examine this issue, we visualized the attention weight, i.e., $\alpha(q(x), k_j)$ in Eq. (4), of each observed point to all target points. These attention weights provide an intuitive description of the contribution by observed points, which also represents the learned neural basis functions as stated in Section 3.4.

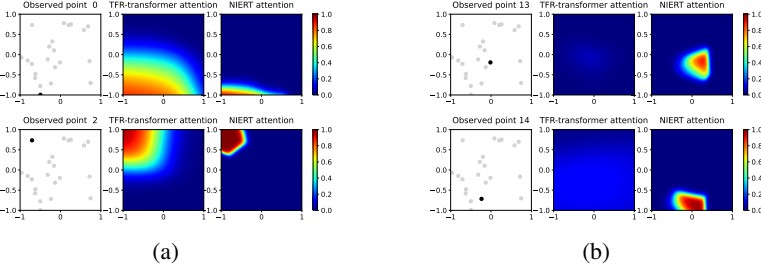

Figure 5: Contributions by observed points for interpolation. The 2D instance is same to that used in Figure 3. We randomly select 4 observed points and extract their attention weights from the final attention layer of NIERT and TFR-transformer. The contributions by other 18 observed points are shown in Section D.2 in the Supplementary material

As shown in Figure 5, when using TFR-transformer, the contributions by observed points are considerably imbalanced: on one side, some observed points (e.g., Fig. 5a) might affect their neighboring target points in a large region; on the other side, the other observed points (e.g., Fig. 5b) have little contributions to interpolation. In contrast, when using NIERT, contributions by an observed point are much more local and thus targeted. More importantly, all observed points have contributions to interpolation. These results demonstrate that NIERT can exploit the correlation between observed points and target points more effectively.

## 4.5 ABLATION STUDY

**The effects of partial self-attention:** For a specific interpolation task, the interpolation function is determined by the observed points only. To investigate the effects of partial self-attention in avoiding interference of target points, we evaluated NIERT on the test sets with various numbers of target points. Here, we compared NIERT with its two variants, one using vanilla self-attention, and the other using partial self-attention with the target points' correlation added. Both of them were trained using the same training sets (the number of target points varies within [206, 246]).

As illustrated by Figure 6, the two NIERT variants show poor performance for the tasks with few target points, say less than 64 target points or more than 768 target points. In contrast, NIERT, which uses partial self-attention, always performs stably without significant changes in accuracy.

The results clearly demonstrate that the partial self-attention mechanism allows NIERT to be free from the unexpected effects of the target points.

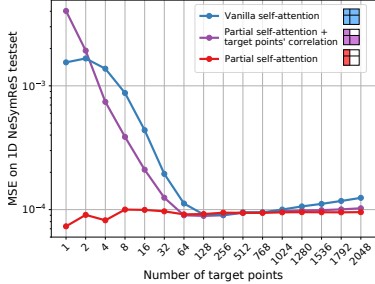

Figure 6: The robustness of NIERT to the number of target points. Here, NIERT using three attention mechanisms trained on the 1D NeSymReS dataset

Figure 7: The convergence of NIERT, NIERT with pre-training, and the existing approaches. Here, these models are trained on TFRD-ADlet dataset

**The effects of pre-training technique:** To investigate the effects of the pre-training technique, we show in Figure 7 the training process of two versions of NIERT, one without the pre-training technique, and the other enhanced with pre-training. As depicted by the figure, even at the first epoch, the pre-trained NIERT shows a sufficiently high interpolation accuracy, which is comparable with the fully-trained BANP and TFR-transformer. Moreover, the performance of the pre-trained NIERT improves at roughly the same convergence speed as the original NIERT. At the final epoch, the pre-trained NIERT decreases the interpolation error to nearly half of that of the original NIERT.

These results clearly suggest that the experience learned by NIERT from the interpolation task in one application field has the potential to be transferred to the tasks in other application fields.

## 5    DISCUSSION AND CONCLUSION

We present in the study an accurate approach to numerical interpolation for scattered data. The specific features of our NIERT approach are highlighted by the full exploitation of the correlation between observed points and target points through unifying scattered data representation. At the same time, the use of the partial self-attention mechanism can effectively avoid the interference of target points onto the observed points. The enhancement with the pre-training technique is another special feature of NIERT. The advantages of NIERT in interpolation accuracy have been clearly demonstrated by experimental results on both synthetic and real-world datasets. Analysis suggests that the advantages of NIERT mainly come from the fact that both observed and target points are embedded into the same feature space at each layer, which makes the message passing from observed points to target points more efficient and informative.

The current version of NIERT has a computational complexity of $O(n(m+n))$, thus cannot handle the interpolation tasks with extremely large amounts of observed points due to the limitations of GPU memory size. Compared with the lightweight traditional methods, our NIERT approach has a larger model with expensive computation to learn complex function distribution, which limits its application in cost-sensitive scenarios.

One possible strategy to improve NIERT's efficiency is sparse partial self-attention: As shown in Figure 5, the contribution by an observed points is quite local and targeted. Thus, for a target point, its neighboring observed points dominate the estimation of its value. Considering the neighborhood only might greatly reduce the computation cost. We will investigate this strategy in future studies.

We expect NIERT, with extensions and modifications, to greatly facilitate numerical interpolations in a wide range of engineering and science fields.

## REPRODUCIBILITY STATEMENT

In the Supplementary material, we provide the anonymous link to the source code for the key experiments. Instructions on how to generate data and train the models are described in detail in the code base. We also provide the construction method or source of each dataset and list the details of experimental configurations. We have fully checked our implementation and verified that our method is reproducible through sufficient experiments.

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

# SUPPLEMENTARY MATERIAL

## TABLE OF CONTENT

## A  DETAILS OF IMPLEMENTATION AND EXPERIMENTS

### A.1  AVAILABILITY OF THE SOURCE CODE OF NIERT

We provide the details of implementation and experiments in this section and deposit source code of NIERT at `https://anonymous.4open.science/r/NIERT-2BCF`.

### A.2  HYPERPARAMETER SETTINGS

Table 5 lists the the hyper-parameters of NIERT when running on the four representative datasets, including NeSymReS, TFRD-ADlet, D30 and PhysioNet datasets. For fair comparison, we set up the TFR-transformer with the same $H$ and $d_{model}$, and the number of its encoder layers is set to $L - 1$ and that of its decoder is set to 1 following (Chen et al., 2021).

| Parameter name | Symbol | Parameter value in experiments on | | | |
|---|---|---|---|---|---|
| | | NeSymReS | TFRD-ADlet | D30 | PhysioNet |
| Number of layers | $L$ | 6 | 3 | 6 | 3 |
| Hidden dimension | $d_{model}$ | 512 | 128 | 128 | 128 |
| Number of heads | $H$ | 8 | 4 | 8 | 4 |
| $x$'s embedding dimension | $d_{xemb}$ | $16{\times}d_x$ | 32 | $16{\times}d_x$ | 16 |
| $y$'s embedding dimension | $d_{yemb}$ | 16 | 16 | 16 | 592 |

Table 5: Hyper-parameters of NIERT in experiments on the four representative dataset

**Training on NeSymReS dataset:** For experiments on NeSymReS dataset, we trained NIERT and the neural approaches for comparison on two NVIDIA GeForce RTX 3090 GPUs for 160 epochs with a batch size of 128. Adam optimizer with a learning rate of $1.0 \times 10^{-4}$ and no schedules are performed for parameters optimization.

**Training on D30 dataset:** For experiments on D30 dataset, NIERT and the neural approaches for comparison are trained on one NVIDIA GeForce RTX 3090 GPUs for 100 epochs with a batch size of 128. Adam optimizer with a learning rate of $1.0 \times 10^{-4}$ and no schedules are performed for parameters optimization.

**Training on TFRD-ADlet and PhysioNet datasets:** For experiments on TFRD-ADlet dataset, we trained NIERT and the approaches for comparison on one NVIDIA GeForce RTX 3090 GPU for 100 epochs with a batch size of 5. Adam optimizer with a learning rate of $5.0 \times 10^{-4}$ with a decay rate of 0.97 is performed. For experiments on PhysioNet dataset, we trained NIERT on one NVIDIA GeForce RTX 3090 GPU for 160 epochs with a batch size of 32. Adam optimizer with a learning rate of $5.0 \times 10^{-4}$ and with a decay rate of 0.97 is performed.

**Pre-training for TFRD-ADlet and PhysioNet datasets:** Pre-trained NIERT on 2D NeSymReS is used to fine-tune on TFRD-ADlet dataset. To deal with the problem that the data distribution is too different between NeSymReS data and temperature field in TFRD-ADlet, we simply normalize the data of temperature field to the value range of 2D NeSymReS data before fine-tuning.

To apply NIERT on PhysioNet dataset in which each instance has so many dependent variables with a number of 37, we increase the number of MLPs for dependent variables to 37 in the embedding layer to embed them separately. The pre-training instances for these tasks are correspondingly different. For convenience, we stack 37 random 1D functions in NeSymReS dataset as an instance for pre-training. Although the data generated using such construction method is considerable different from PhysioNet data, we still observe that pre-training technique improves the interpolation accuracy as shown in Table 4.

### A.3  TRAINING ALGORITHM

The training process is depicted in Algorithm 1. Specifically, we prepare the training interpolation tasks by first sampling functions from a distribution $\mathcal{F}$ and then sampling observed points $O$ and target points $T$ on each function. When training NIERT, we set the loss function as the error between

---

**Algorithm 1** NIERT training process

---

**Require:** NIERT model $M_\theta$ with parameters $\theta$, epoch number $N$, batch size $B$, domain $X$, function distribution $\mathcal{F}$ and error metric function $\text{Error}(\cdot, \cdot)$

**for** $k$ in $\{1..N\}$ **do**
    $J \leftarrow 0$
    **for** $b$ in $\{1..B\}$ **do**
        $f, \{x_i\}_i \leftarrow$ sample a function and scatter points from $\mathcal{F}, X$
        $\{y_i\}_i \leftarrow$ calculate $f$ on $\{x_i\}_i$
        $O, T \leftarrow$ split and mask scatter points with values $\{(x_i, y_i)\}_i$
        $\{\hat{y_i}\}_i \leftarrow M_\theta(O, T)$
        $J \leftarrow J + \sum_i \text{Error}(\hat{y_i}, y_i)$
    **end for**
    Compute the gradient $\nabla_\theta J$ and update $\theta$
**end for**

---

the estimated values and the corresponding ground-truth. Note that errors acquired on both observed points and target points are accounted into loss function.

## B DETAILS OF DATASETS

We evaluated NIERT approach using four datasets, including synthetic datasets NeSymReS, TFRD-ADlet and D30, and a real-world dataset PhysioNet. We first clearly list the statistics of interpolation tasks of these datasets and then introduce the datasets in detail in the following subsections.

The statistical information of the interpolation tasks used for training in each dataset is shown in Table 6. In practice, for each instance in the training set, i.e., $N$ scattered points from a certain function, we randomly select $n$ of them as observed points and the remaining $N - n$ as target points. For the NeSymReS dataset, we randomly select a fixed range of numbers as the number of observed points (e.g. [5,25] for 1D NeSymReS dataset). For the real-world PhysioNet dataset, which has a variable number of scattered points for its instances, distributed in the range of $[20, 190]$, we therefore select observed points from all points at a fixed rate (say 50%,70% or 90%).

| Dataset | Subset tag | #All points $N$ | #Observed points $n$ | #Target points $m$ |
|---|---|---|---|---|
| NeSymReS | 1D | 256 | [10,50] | $N - n$ |
| | 2D/3D/4D | 512 | [10,50] | $N - n$ |
| TFR-ADlet | - | 40000 | 37 | 39963 |
| D30 | 10D/20D/30D | 256 | 64 | 192 |
| PhysioNet | 50% | [20,190] | $0.5 \times N$ | $N - n$ |
| | 70% | [20,190] | $0.7 \times N$ | $N - n$ |
| | 90% | [20,190] | $0.9 \times N$ | $N - n$ |

Table 6: Statistics of the interpolation tasks used for training in each dataset

### B.1 SYNTHETIC DATASET I: NESYMRES

We used the NeSymReS[1] dataset to test the performance of NIERT for mathematical function interpolation. This dataset was constructed following Biggio et al. (2021) with only slight modification for the sake of numerical interpolation. For clarity, we put the details of construction procedure in the subsection B.1.1 below, and list the summarized characteristics of this dataset as follows.

**Instance construction:** We randomly sampled $N$ points from $X = [-1, 1]^{d_x}$ ($d_x$ represents the point dimensionality) and calculate the values for these points with a mathematical function $f$. From these points, we randomly picked $n$ points as observed points and used the left-over as target points. In the study, we chose different $d_x$ for constructing datasets, including 1D, 2D, 3D, and 4D. For 1D NeSymReS dataset, we set the scattered points number $N$ as 256. For 2D/3D/4D NeSymReS dataset, we set $N$ as 512. We control observed points number $n$ within the range of $[5, 50]$.

**Training set and test set:** Each instance in training or test set was built using a mathematical function $f$. These functions were generated using a function sampler as performed in (Biggio et al., 2021). During the training process, 1 million instances are dynamically sampled at each epoch as training set. We used the $L_2$-form loss function for this dataset. We also generated 12000 instances and used them as test set.

**Dimensionality of points:** We evaluated NEIRT using scatter data with various dimensionality, including 1D, 2D, 3D, and 4D.

**Approaches for comparison:** We compared NIERT with five representative interpolation approaches, including radial basis function (RBF) (Powell, 1987; Fornberg & Zuev, 2007), MIR (Wackernagel, 2003), conditional neural process (CNP)(Garnelo et al., 2018), attentive neural process (ANP)(Kim et al., 2019) and TFR-transformer (Chen et al., 2021). Among these approaches, RBF and MIR are classical approaches that use explicit basis functions, while CNP, ANP and TFR-transformer use neural networks to learn how to interpolate.

---

[1]Built based on https://github.com/SymposiumOrganization/NeuralSymbolicRegressionThatScales.

### B.1.1 NeSymReS dataset construction

Following the study of Biggio et al. (2021), firstly, we generate equation *skeletons* which is refer to the symbolic equation where numerical constants are replaced by placeholders (Biggio et al., 2021). Each equation skeleton has the configured number ($d_x$) of independent variables symbols. For example, $y = \sin(C_1 x_1) + C_2 x_2^2$ is a possible generated equation skeleton which includes constants placeholders $C_1$ and $C_2$ and two variables $x_1$ and $x_2$. Such equation skeletons are considered expression trees during generation. Each randomly-generated expression tree has up-to 5 non-leaf nodes, i.e, function operators. Unnormalized weighted distribution shown in Table 7 is used for sampling each non-leaf node. Each leaf node has a probability of 0.8 of being an independent variable and 0.2 of being an integer. Different from (Biggio et al., 2021) using almost all elementary functions symbols including discontinuous ones like ln, arcsin, tan for symbolic regression tasks, we only use the operators listed in Table 7 which guarantee that the generated function is continuous in the whole domain $X = [-1, 1]^{d_x}$, to make it more suitable for interpolation tasks.

| Operator | + | × | − | .2 | .3 | exp | sin | cos |
|---|---|---|---|---|---|---|---|---|
| Unnormalized probability | 10 | 10 | 5 | 4 | 2 | 4 | 4 | 4 |

Table 7: Operator and corresponding un-normalized probabilities during the generating process of mathematical functions in NeSymReS dataset

Secondly, constants values are independently sampled from a uniform distribution $\mathcal{U}(1, 5)$ and we get a set of completely-defined mathematical-expression functions. Then we normalize those functions to make their values range in $[0, 1]$ and multiply it with a random number range $[0.9, 1]$ to produce diversity.

After those completely-defined functions obtained, we sample interpolation task for each function by randomly sampling a set of support points $\{x_i\}_i$ in $X$, evaluating the function and get the corresponding $\{y_i\}_i$, and then splitting those data points into a observed points set and a target points set. 512 scatter points are sampled using each function, of which a random number (ranging in $[5, 50]$) of points are set up as the observed points and the rest are set up as the target points.

During the generation of interpolation tasks, results with invalid values (NaN or inf) are removed. For $d_x$ in configurations $\{1, 2, 3, 4\}$, we generate datasets separately and use them for training and testing. In particular, when $d_x$ is configured, we generate 150 equation skeletons set for testing and 1 million skeletons set for training. Each skeleton in the training set existing in the test set has been removed. At training time, interpolation tasks are sampled using equation skeletons from the training set in real time. At testing time, we use an interpolation task set containing 10000 instances, which are generated by the testing skeletons set.

### B.2 Synthetic dataset II: TFRD-ADlet

TFRD-ADlet[2] (Chen et al., 2021) is a synthetic dataset for 2D temperature field reconstruction where each instance represents a simulated 2D temperature field containing several heat source components and a specific Dirichlet conditioned boundary. The goal of each task instance is to reconstruct the whole temperature field according to a limited number of observed points with measured temperature. We use the TFRD-ADlet dataset to test the performance of NIERT for 2D temperature field reconstruction. The characteristics of this dataset are summarized as follows.

**Instance construction:** Each instance has $200 \times 200$ regular grid points that represent the temperature field in a $0.1m \times 0.1m$ square area. Among these grid points, 32 points have their temperate known and used as observed points. The other 3968 points are used as target points.

**Training set and test set:** We have a total of 10,000 training instances and 10,000 test instances. For the sake of fair comparison, we also use $L_1$-form loss function for this dataset as performed in Ref. (Chen et al., 2021).

**Dimensionality of points:** The grid points are in a 2D plane.

---

[2]TFRD-Alet is downloadable at https://pan.baidu.com/s/14BipTer1fkilbRjrQNbKiQ, password: 'tfrd'.

**Approaches for comparison:** For this dataset, we compared NIERT with three neural network-based interpolation approaches, including conditional neural process (CNP)(Garnelo et al., 2018), attentive neural process (ANP)(Kim et al., 2019) and TFR-transformer (Chen et al., 2021). We also compared NIERT with its enhanced version that was pre-trained using the NeSymReS dataset.

### B.3 SYNTHETIC DATASET III: D30

In order to verify the scalability of NIERT on higher dimensional data, we specially constructed a synthetic dataset (called D30 dataset) containing three subsets whose dimensionalities are 10, 20 and 30 respectively.

**Instance construction:** Each function for sampling one interpolation task is obtained by the summation of $K$ randomly sampled $d_x$-dimensional Gaussian functions, which can be formalized as

$$f(x) = \sum_{k=1}^{K} A_k \exp\left(-\frac{1}{2}\frac{(x-c_k)^2}{\sigma_k^2}\right).$$

We fix $K$ as 5. For each Gaussian function using in each function, we uniformity sampled the center $c_k$ from $[-1,1]^{d_x}$, weight $A_k$ from $[-1,1]$ and width $\sigma_k$ from $[\sigma_{d_x}, 2\sigma_{d_x}]$ where $\sigma_{d_x}$ is configured as 1, 2 and 4 when $d_x$ is 10, 20 and 30 respectively.

**Training set and test set:** For each dimensionality of 10, 20 or 30, we created a training set containing 256K instances and a test set containing 512 cases from this function distribution. Each instance includes 64 observed points and 192 target points, which are uniformly sampled from $[-1,1]^{d_x}$.

**Dimensionality of points:** The data points in three subsets have the dimension of 10, 20 and 30 respectively.

**Approaches for comparison:** On D30, we compared NIERT with five representative interpolation approaches, including radial basis function (RBF) (Powell, 1987; Fornberg & Zuev, 2007), MIR (Wackernagel, 2003), conditional neural process (CNP)(Garnelo et al., 2018), attentive neural process (ANP)(Kim et al., 2019) and TFR-transformer (Chen et al., 2021).

### B.4 REAL-WORLD DATASET: PHYSIONET

PhysioNet[3], excerpted from the PhysioNet Challenge 2012 (Silva et al., 2012), is a real world dataset collected from intensive care unit (ICU) records for time-series data interpolation. Each point in an instance represents a measurement at a specific time, where each measurement contains up to 37 physiological indices. It should be pointed out that this dataset is a representative of hard interpolation tasks due to the sparsity and irregularity of the records.

**Instance construction:** Each instance consists of multiple points, each of which represents a measurement of a patient at a specific time. Following the study (Shukla & Marlin, 2021), we randomly divided the points into observed points and target points. We also set the ratio of observed points at five levels, i.e., 50%, 70%, and 90%, and trained and evaluated NIERT using the thus-acquired datasets.

**Training set and test set:** We randomly divided the 8,000 instances acquired from the PhysioNet Challenge 2012 into training set and test set with a ratio of 4:1. For the sake of fair comparison, we use $L_2$-form loss function for this dataset as performed in (Shukla & Marlin, 2021).

**Dinsionality of points:** Each point in an instance represents a measurement at a specific time and each measurement contains up to 37 physiological indices; thus, the independent variable $x$ is 1D while the dependent variable $y$ has a dimensionality of 37.

**Approaches for comparison:** On this dataset, we compared NIERT with four representative approaches designed for time-series data interpolation, including RNN-VAE (Chung et al., 2014), L-ODE-RNN (Chen et al., 2018), L-ODE-ODE(Rubanova et al., 2019), and mTAND-Full(Shukla & Marlin, 2021). The details of these approaches are provided in Section C. We also compared NIERT with its enhanced version that was pre-trained using the NeSymReS dataset.

---

[3]PhysioNet is downloadable at https://physionet.org/content/challenge-2012/1.0.0/.

## C   APPROACHES FOR COMPARISON

**Radial basis function (RBF)**   RBF is one of the most commonly-used scattered data interpolation methods. It adopts a specific type of radial basis functions on observed points and uses their linear combination to represent the target function. We use the RBF interpolation implementation in SciPy (Virtanen et al., 2020) and multiquadric function as basis function type for the experiments.

**MIR**   MIR [4] is another multivariate interpolation and regression method for scattered data sets proposed by (Wang et al., 2010). MIR represents the approximation error with Taylor expansions and minimizes the approximation error to find the basis functions.

**Conditional nerual process (CNP)**   CNP proposed by (Garnelo et al., 2018) is an neural model able to learn to predict distributions of target points values given a series of observed points. In order to fully verify the accuracy of interpolation, we let CNP to predict the values of target only and the training criteria function is set to be the prediction error of values of target points in the experiments. We borrowed the parameter settings from the comparison experiments in the work of Chen et al. (2021), where the intermediate layer dimensions of the MLP encoder are $[d_x + d_y, 128, 128, 256, 512]$, and these of MLP decoder are $[512 + d_x, 256, 256, 128, 128, d_y]$.

**Attentive nerual process (ANP)**   ANP (Kim et al., 2019) leverages attention mechanism in CNP and improves the prediction performance. In the experiments, the criteria function are fixed as same as CNP above. We borrowed the parameter settings from Chen et al. (2021), where the hidden dimension is 512, attention head number is 4, and the encoder and decoder have two attention layers respectively.

**Bootstrapping attentive nerual process (BANP)**   BANP (Lee et al., 2020b) employs bootstrap technique to further improve the performance of ANP. In the experiments, the criteria function are fixed as same as CNP and ANP above. We set its hyperparameters to be the same as ANP, except for its unique hyperparameter, i.e., the number of samples, which is set to 2 for computational cost considerations.

**TFR-Transformer**   Transformer (Kim et al., 2019) is originally proposed to solve tasks in natural language processing. Chen et al. (2021) adopts Transformer in 2-dimensional temperature field reconstruction using scattered observations. Compared with vanilla transformer, TFR-Transformer removes positional encoding, encodes the observations using encoder, using cross-attention mechanism between observations' encoding and target points to represents targets' features at decoder, and using a MLP to map targets' features to values. We set its hidden space dimension, the number of attention layers and the number of attention heads in each experiment to be the same as our method NIERT. These hyperparameters are listed in Table 5.

**RNN-VAE**   RNN-VAE is a VAE-based model where the encoder and decoder are standard RNN models. Gated Recurrent Unit (GRU) (Chung et al., 2014) module is configured as the recurrent network.

**L-ODE-RNN**   L-ODE-RNN refers to latent neural ODE model where the encoder is an RNN and decoder is a neural ODE proposed in (Chen et al., 2018).

**L-ODE-ODE**   Rubanova et al. (2019) proposes ODE-RNN model which generalize RNNs to have continuous-time hidden dynamics defined by ODEs. L-ODE-ODE refers to the model where the encoder is an ODE-RNN and decoder is a neural ODE.

**mTAND-Full**   Shukla & Marlin (2021) proposes mTAND-Full for interpolation and classification of sparse, irregularly sampled, and multivariate time series data. mTAND-Full performs time attention mechanism to learn temporal similarity and Bidirectional RNNs to encode temporal features. Mask mechanism makes the representation of missing data and target points convenient and easy to parallel. For the experiments of RNN-VAE, L-ODE-RNN, L-ODE-ODE, and mTAND-Full approaches, we use the same hyperparameter settings as Shukla & Marlin (2021).

---

[4]MIR's implementation can be found at http://web.mit.edu/qiqi/www/mir/.

# D ADDITIONAL EXPERIMENTAL RESULTS

## D.1 ADDITIONAL CASE STUDIES

**Cases from 1D & 2D NeSymReS test set**

In each example of 1D interpolation task extracted from NeSymReS test set, the blue curve represents the ground-truth function while the red curves represent the interpolation functions reported by NIERT and the existing approaches.

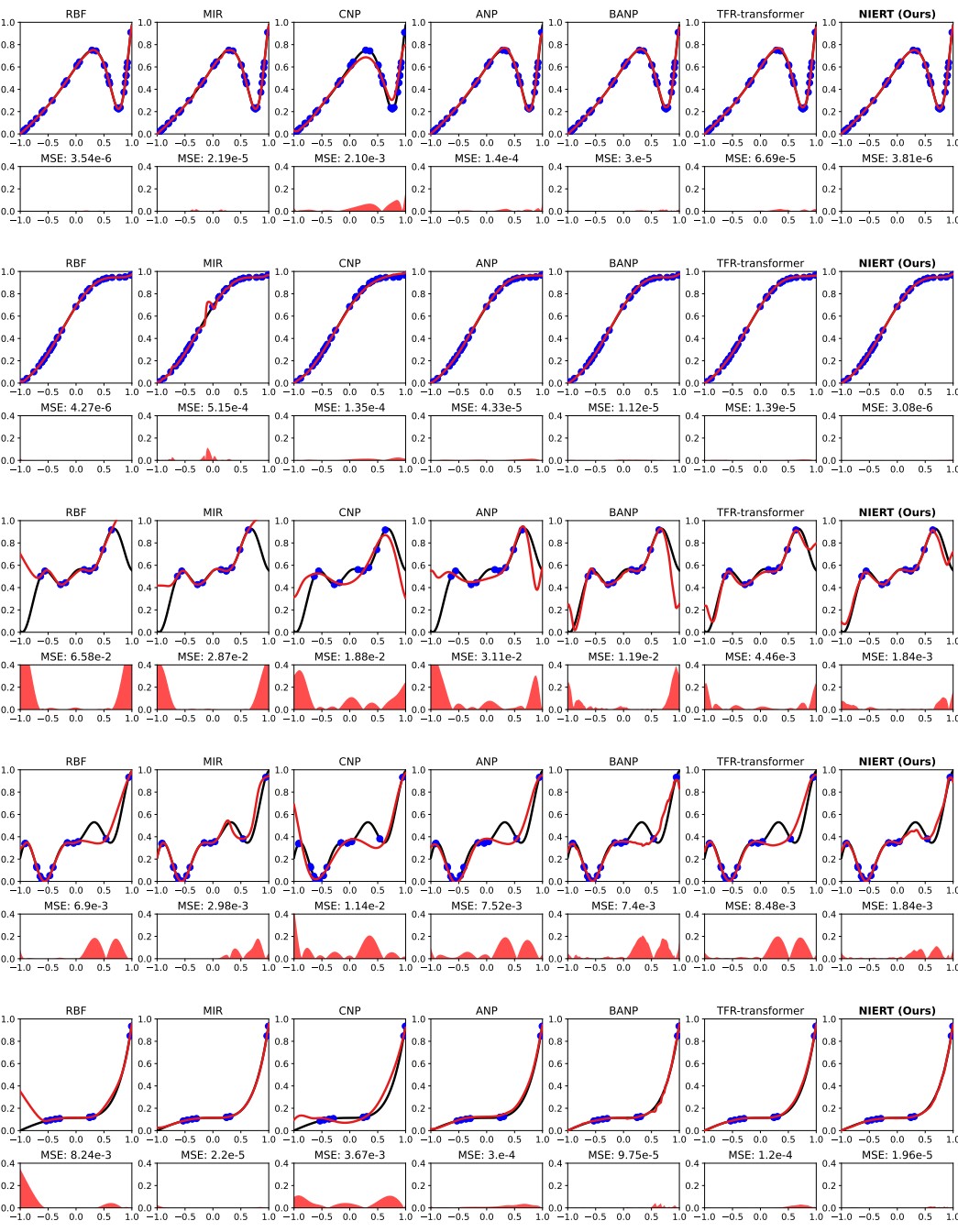

Figure 8: Additional cases from 1D NeSymReS test set

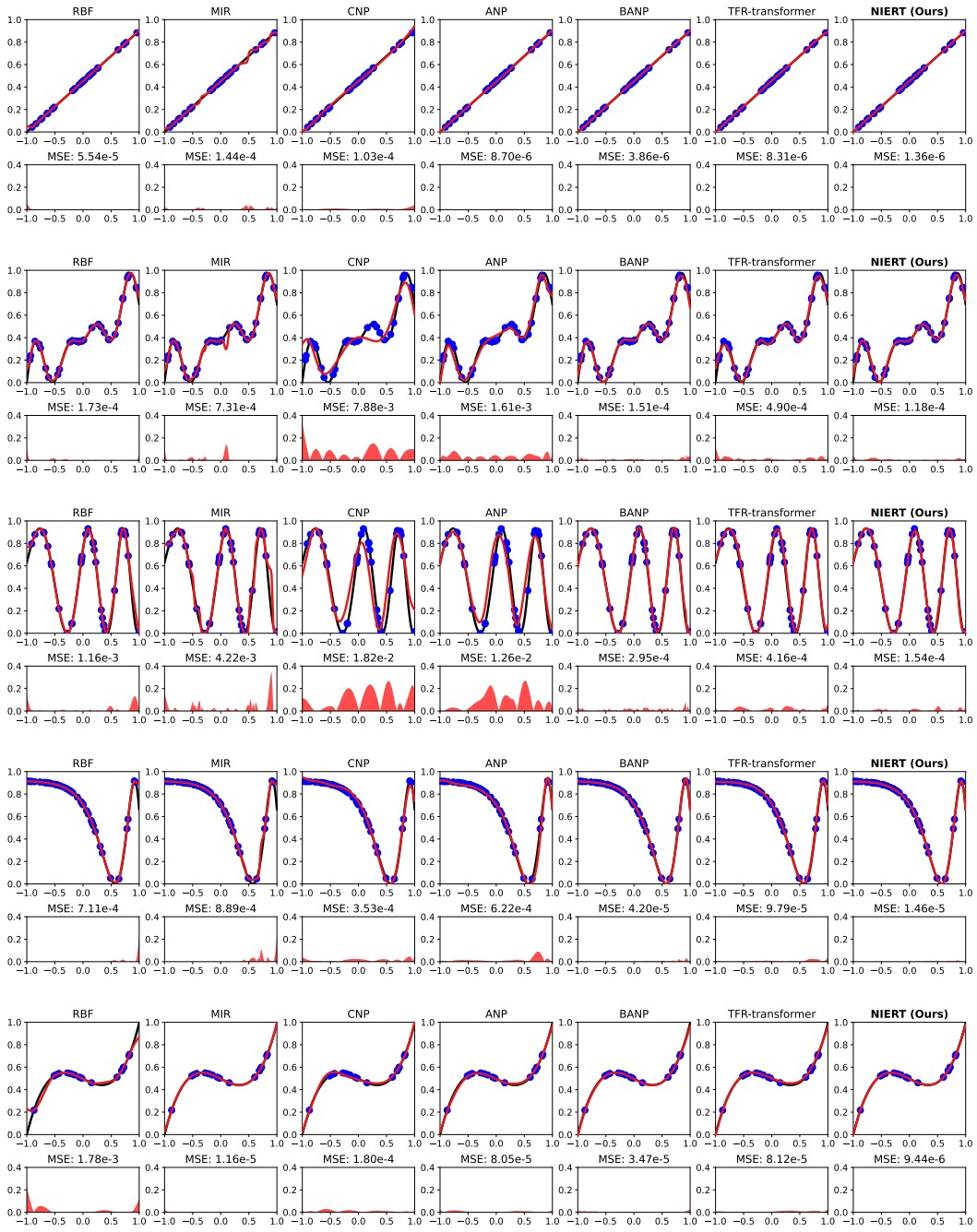

Figure 9: Additional cases from 1D NeSymReS test set

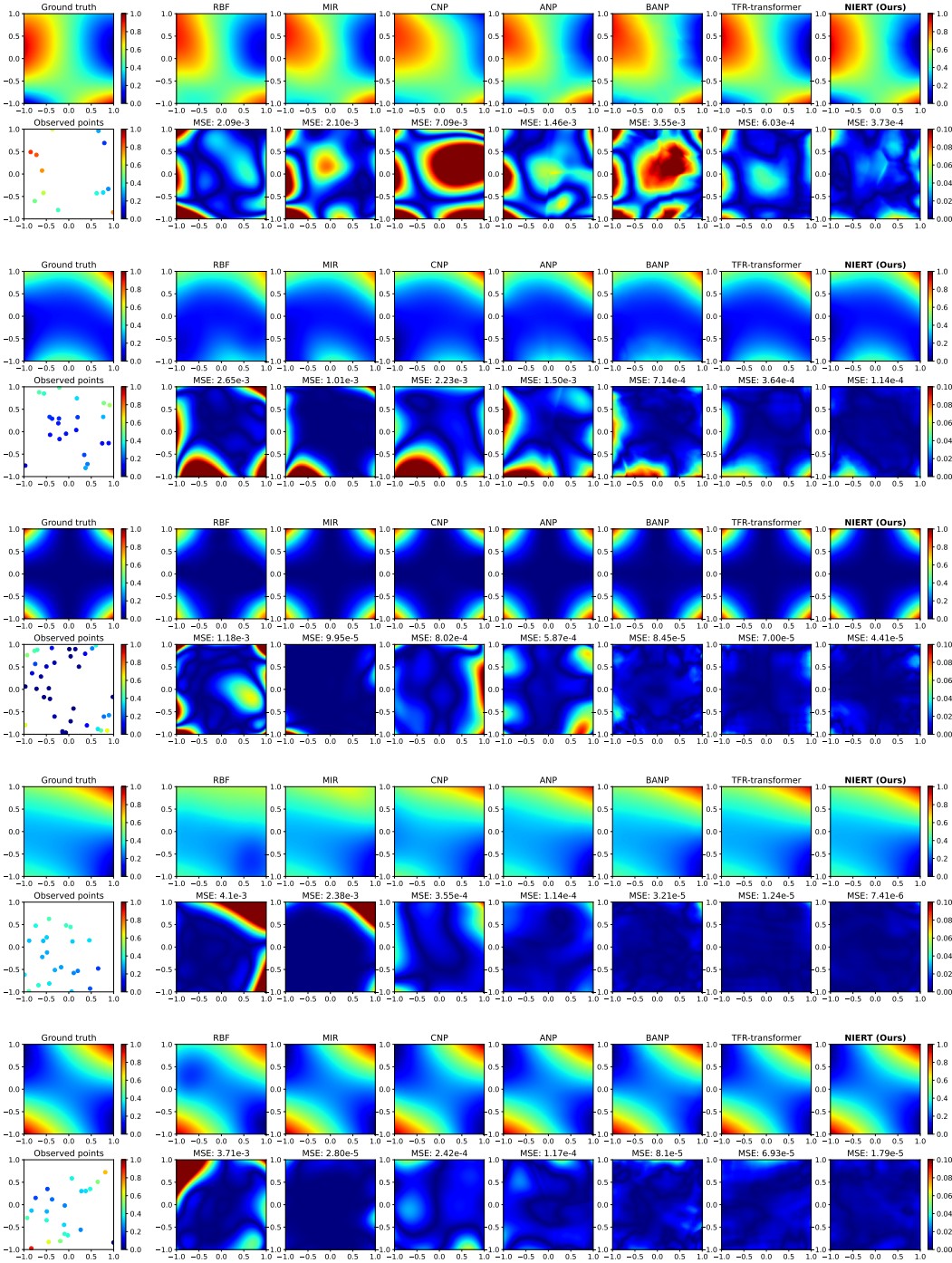

Figure 10: Additional cases from 2D NeSymReS test set

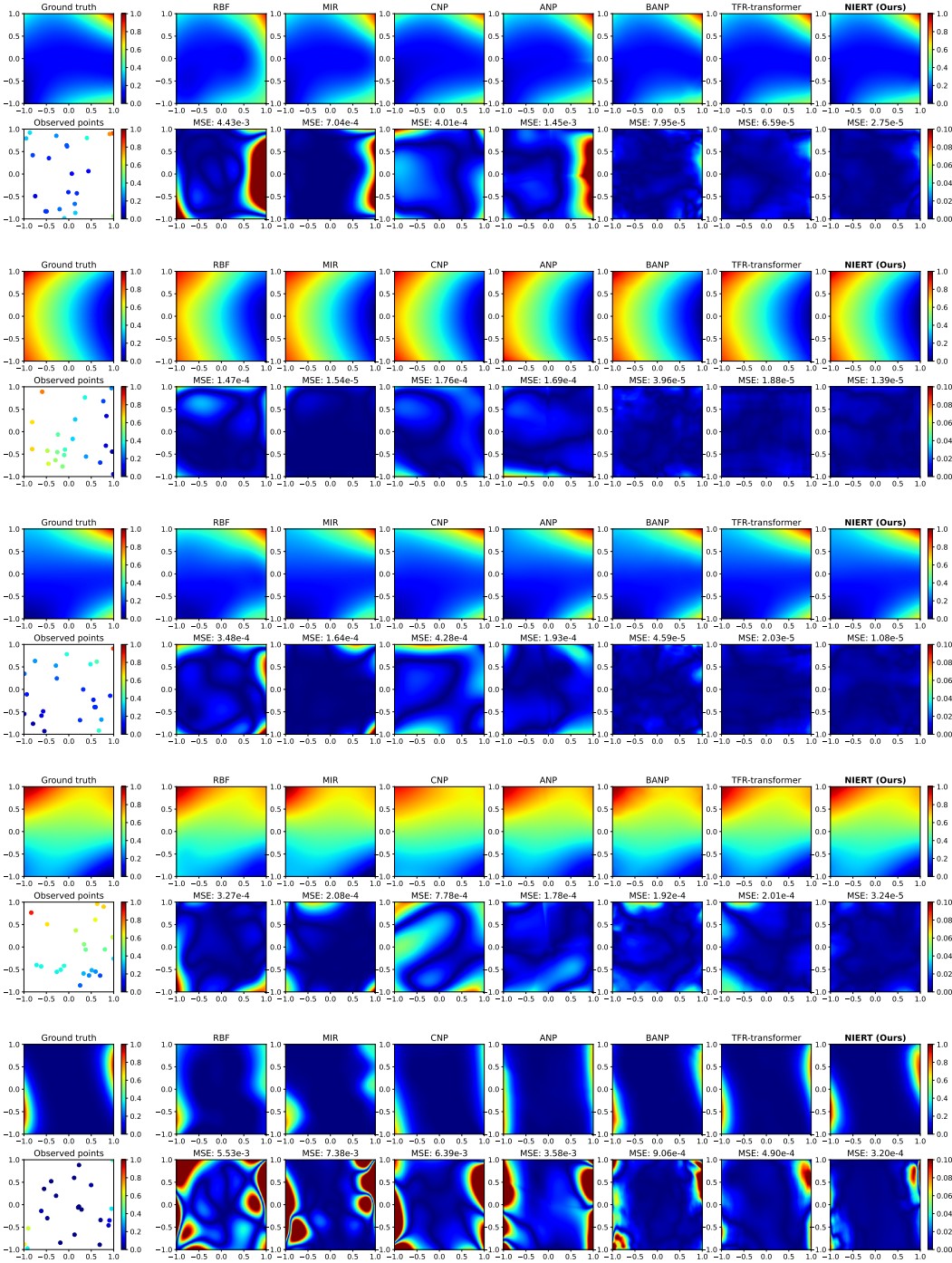

Figure 11: Additional cases from 2D NeSymReS test set

**Cases from TFRD-ADlet**

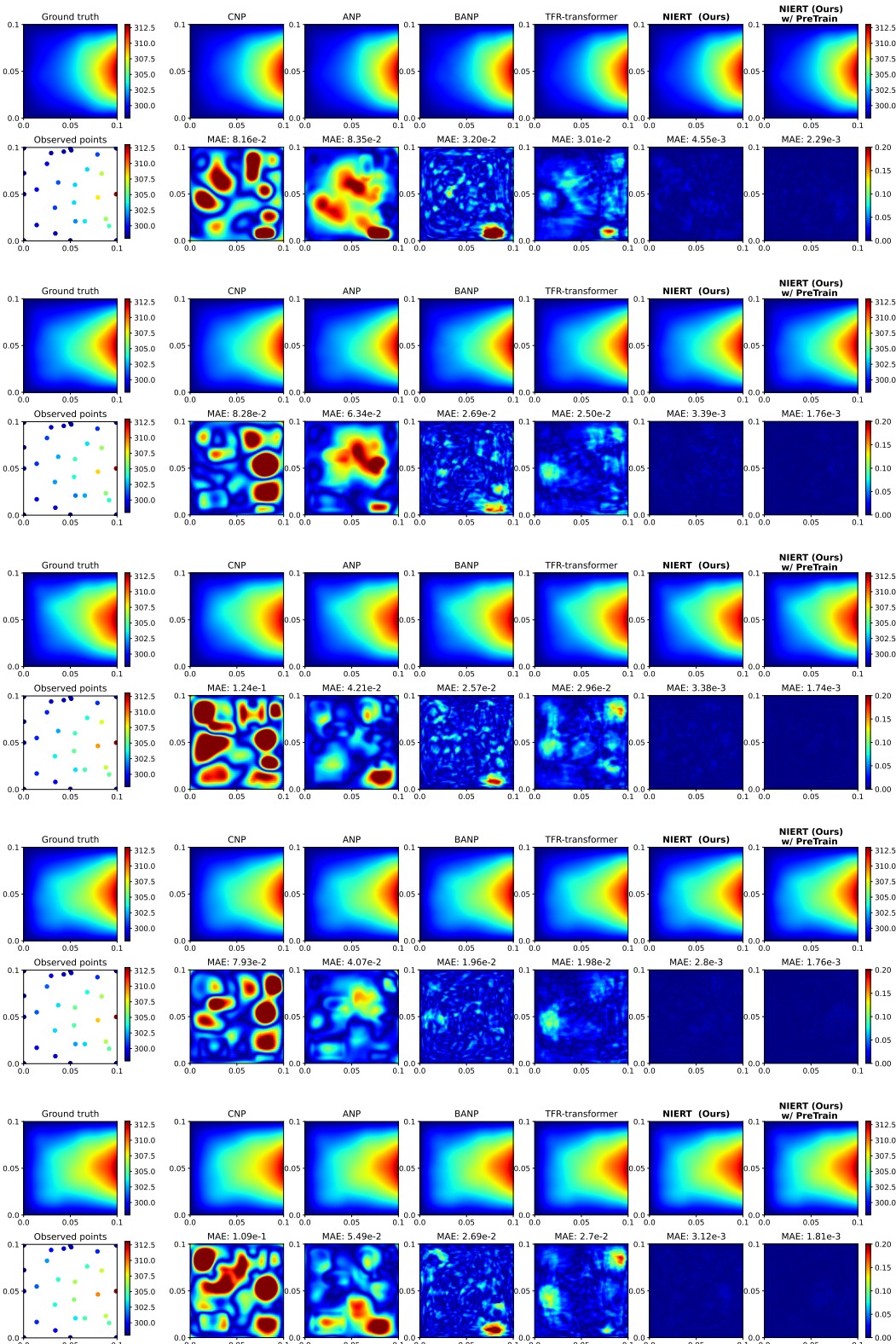

Figure 12: Additional cases from TFRD-ADlet test set

## D.2 CONTRIBUTION ANALYSIS OF OBSERVED POINTS FOR INTERPOLATION

As supplements to Figure 6, all observed points' attention weights extracted from the final attention layer of NIERT and TFR-transformer are visualized in Figure 13.

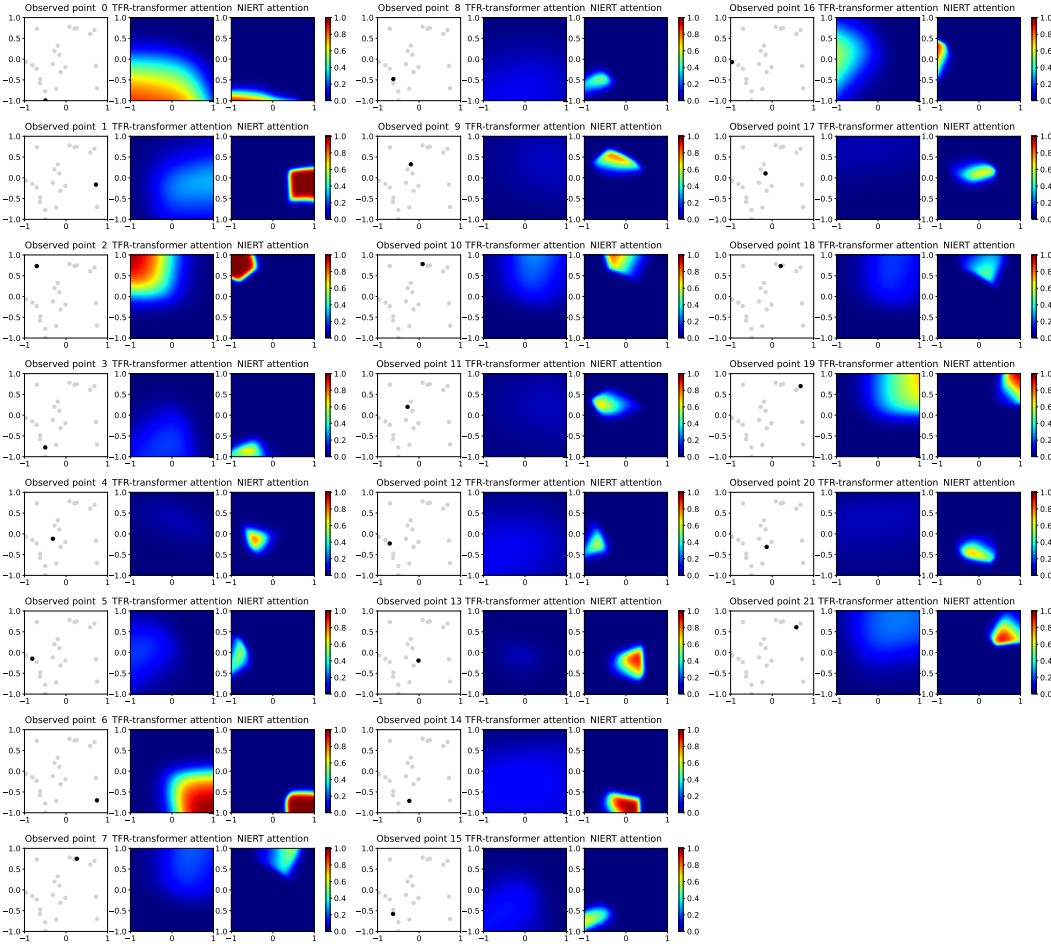

Figure 13: All observed points and the corresponding extracted attention response

The contributions by observed points are quit imbalanced using TFR-transformer. In particular, the intensity and area of the contributions of the five observed points 0,2,6,16,19 cover almost the entire domain, while the remaining 17 observed points only produce negligible contribution. In contrast, when using NIERT, contributions by an observed point are much more local and thus targeted and all observed points have contributions to interpolation. This shows that NIERT can fully exploit the relationship between observed points and target points.

## D.3 ADDITIONAL ABLATION STUDIES

**The effects of different model depths**

We carried out experiments on 2D NeSymReS dataset using NIERT with hyper-parameter $L$ varying from 3 to 7. Then evaluate the models on the 2D NeSymReS test dataset. Accuracy are listed in below Table 8. The results show that NIERT with 7 transformer layers has the best accuracy on the test set, and NIERT with 6 transformer layers has reached a comparable level. Therefore, in the experiments on NeSymReS data set, we use $L = 6$ to balance efficiency and accuracy.

**The effects of different hidden dimensions**

| Interpolation approach | Number of transformer layers $L$ | | | | |
|---|---|---|---|---|---|
| | 3 | 4 | 5 | 6 | 7 |
| NIERT | 66.812 | 60.133 | 52.098 | 45.319 | **44.043** |

Table 8: The interpolation accuracy of NIERT (MSE $\times 10^{-5}$) under various settings of transformer layer number $L$ on 2D NeSymReS dataset

We also carried out experiments on 2D NeSymReS dataset using NIERT with smaller hidden dimensions $d_{model}$, say from 256, 128 and 64. Then evaluate the models on the 2D NeSymReS test dataset. Accuracy are listed in below Table 9. The results show that when the hidden dimension is within 512, NIERT's interpolation accuracy is higher when the hidden dimension is larger.

In this experiment, we fix other super parameters, say model depth $L$ as 6 and number of heads as 8.

| Interpolation approach | Different hidden dimensions $d_{model}$ | | | |
|---|---|---|---|---|
| | 64 | 128 | 256 | 512 |
| NIERT | 106.193 | 72.107 | 51.153 | **45.319** |

Table 9: The interpolation accuracy of NIERT (MSE $\times 10^{-5}$) under various settings of hidden dimension $d_{model}$ on 2D NeSymReS dataset

**The effects of prediction error of observed points in loss function**

To verify the contribution of re-predicting values of the observed points to the interpolation task, we conducted an experiment that puts the prediction error of observed points in the loss function, i.e. only minimizes the estimation error of target points. The results are shown in Table 10 which demonstrates that only minimizing estimation error of target points make NIERT performing poorly. This indicates that re-predicting the value of observed points helps NIERT to predict the value of target points more accurately.

| Loss contains prediction error of | Only target points | All points |
|---|---|---|
| MSE ($\times 10^{-5}$) | 48.931 | **45.319** |

Table 10: Interpolation accuracy (MSE $\times 10^{-5}$) of NIERT trained with loss only containing the prediction error of the target points

### D.4 PREDICTION ACCURACY GAP BETWEEN OBSERVED POINTS AND TARGET POINTS

| Prediction accuracy on | MSE ($\times 10^{-5}$) on NeSymReS test sets | | | |
|---|---|---|---|---|
| | 1D | 2D | 3D | 4D |
| Observed points | 1.301 | 4.862 | 3.775 | 2.662 |
| Target points | 8.964 | 45.319 | 77.664 | 72.025 |

Table 11: Prediction accuracy gap between observed points and target points on NeSymReS test set

We carried out an extra experiment for prediction accuracy analysis on observed points and target points. Figure 11 shows that on the observed points, the MSE of prediction is relatively smaller when compared with it of target points as expected.

### D.5 COMPUTATIONAL TRADE-OFF

Our work focuses on improving the accuracy of scattered data interpolation. Considering the practical scenario of efficient computational interpolation, we compare and analyze the accuracy-

computation time trade-off between our approach and previous approaches. We collected the average computation time and MSE per interpolation task for all approaches on the 2D NeSymReS testset, which contains 10,000 test interpolation tasks, and visualize the results in Figure 14. Figure 14, the lower the method the higher the accuracy and the more left the method the more efficient the calculation.

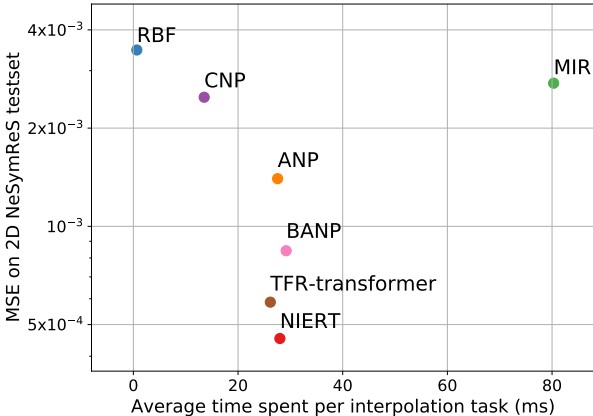

Figure 14: Computational trade-off comparison on 2D NeSymReS testset

From Figure 14, we can see that RBF is the most computationally efficient, but it is also the least accurate. Our approach NIERT has the highest computational accuracy, and its computational efficiency is only slightly reduced compared to the next best approach, TFR-transformer.

It should be noted that the traditional interpolation approach RBF and MIR are tested using Intel(R) Xeon(R) Gold 5218R CPU @ 2.10GHz with 40 cores and the data-driven approaches including CNP, ANP, BANP, TFR-transformer, and NIERT were evaluated on an NVIDIA GeForce RTX 3090 GPU. Here we set the batch size to 1 for evaluation to keep the comparison fair.

For the training times of these different methods, we collected and put them into the Table 12. We used two NVIDIA GeForce RTX 3090 GPUs for training. On a 2D NeSymReS training set dataset containing one million functions, 160 epochs of training were performed. It should be noted that traditional interpolation methods such as RBF and MIR do not require training.

| Interpolation approach | Training time (h) |
|:---:|:---:|
| RBF | - |
| MIR | - |
| CNP | 14.9 |
| ANP | 18.3 |
| BANP | 21.1 |
| TFR-transformer | 17.5 |
| NIERT | 19.1 |

Table 12: Training time comparison on 2D NeSymReS training set

Table 12 indicates that the training times of these data-driven approaches do not differ much. Among them, CNP has the shortest training time (14.9 h) and BANP has the longest training time (21.1 h). The training time of our approach NIERT is 19.1 hours.

