# OpenReview forum: "NIERT: Accurate Numerical Interpolation through Unifying Scattered Data Representations using Transformer Encoder"
_ICLR.cc/2023/Conference — Submitted to ICLR 2023_

### Official Review · Reviewer_uN6v · 2022-10-24

**Confidence:** 2
**Correctness:** 3
**Technical Novelty And Significance:** 3
**Empirical Novelty And Significance:** 3
**Recommendation:** 5

**Clarity, Quality, Novelty And Reproducibility:**

* Clarity: The paper is well-written and easy to follow.
* Quality: The proposed framework performs well in practice.
* Novelty: Based on the claims of the authors, this is the first work that applies deep learning (Transformer) to the numerical interpolation problem and the partial self-attention mechanism is new.
*  Reproducibility: The code is submitted. The experimental settings are reported in detail.

**Strength And Weaknesses:**

## Strength
* The proposed framework exploits the power of Transform in the numerical interpolation where traditional interpolation approaches for scattered data use explicitly pre-defined basis functions to construct interpolation functions.
* The partial self-attention mechanism is new to the best of my knowledge.
* The authors also discuss the connection between NIERT and the traditional approach by showing that partial self-attention is a general form of the RBF interpolation function.
* The proposed framework beats all baselines on all datasets.

## Weaknesses
* The computational trade-off should be discussed e.g, the training time, and the interpolation time.
* The authors should discuss the statistics about the number of observed and target points in each dataset. I wonder how the proposed framework performs in different settings of sequence lengths.
* The authors only focus on the mean absolute error. There is no downstream task for showing that NIERT is better than previous frameworks.

**Summary Of The Paper:**

The paper introduces NIERT which is a new framework for numerical interpolation for scattered data. The key idea of the framework is to use Transformers as a representation encoder and treats observed and target points in a unified fashion by embedding them into the same representation space. In the paper, the authors also propose a new partial self-attention mechanism that could escape from the unexpected interference of target points on observed points. Combining the partial self-attention mechanism,  masking mechanism, and pre-training technique, NIERT outperforms the existing approaches on both real-world and synthetic datasets. For example, NIERT gives the best mean absolute error on the NeSymReS dataset, the TFRD-ADlet dataset, the D30 dataset, and the PhysioNet. The authors also conduct ablation studies to understand the role of partial self-attention and the pre-training technique.

**Summary Of The Review:**

The paper proposes the partial self-attention mechanism and applies it to the numerical interpolation problem. The framework achieves great prediction results however the performance in downstream tasks is missing.

---

> ### Author Response · Authors · 2022-11-11
> **Response to Reviewer uN6v**
>
> We thank the reviewer for his/her approval of our work. The reviewer also raised several concerns which we addressed below.
>
> > **1.** The computational trade-off should be discussed e.g, the training time, and the interpolation time.
>
> We completely agree with the reviewer's comments. We have analyzed and visualized the interpolation time-accuracy tradeoff, and the training time of these approaches on the 2D NeSymReS dataset as a representative. We have put the results in the Supplementary Text (Figure 14 of D.5, Table 12 of D.5).
>
> These results show that our approach NIERT achieves the best interpolation accuracy and its average interpolation time (27.99ms) is not much worse than that of the second-best approach (26.17ms).
>
> These results also show that the training times of these data-driven approaches do not differ significantly on the 2D NeSymReS dataset. Among them, CNP has the shortest training time (14.9 h) and BANP has the longest training time (21.1 h). The training time of our approach NIERT is 19.1 hours.
>
> > **2.** The authors should discuss the statistics about the number of observed and target points in each dataset. I wonder how the proposed framework performs in different settings of sequence lengths.
>
> We completely agree with the reviewer's comments. We have provided statistics on the number of observed and target points for each dataset in the Supplementary Text (Table 6 of Part B).
>
> Evaluated on the interpolation task for a specific dataset, the experimental results show that the interpolation accuracy of our approach NIERT increases with the number of observed points (Figure 2 in Section 4.2). The experimental results also show that the number of observation points does not affect the accuracy of the interpolation of our method (Figure 5 in Section 4.5).
>
>
> > **3.** The authors only focus on the mean absolute error. There is no downstream task for showing that NIERT is better than previous frameworks.
>
> We understand the reviewers' concern that our work lacks downstream tasks.
>
> In fact, unlike approaches such as BERT in NLP, where the main model only learns the representation of text, interpolation learning of scattered data itself is capable of solving a large class of practical tasks, including temperature field reconstruction, interpolation of irregularly sampled time series, etc. Such interpolation tasks always use MSE or MAE directly as an evaluation metric for their accuracy. Therefore, we believe that our practice, i.e., evaluating interpolation accuracy with MSE or MAE on the four representative data sets, is appropriate and sufficient.
>
> It is undeniable that numerical interpolation has many downstream tasks in various fields, such as the numerical solution of partial differential equations. Evaluating our approach to these downstream tasks is of importance, but beyond the scope of this work. We leave it for future work.

---

> > ### Comment · Reviewer_uN6v · 2022-11-17
> > **Response to the authors**
> >
> > Thank you for your response. All my questions have been addressed. I will adjust my score based on other discussions due to my lack of knowledge of the literature.
> >
> > Best,

---

> > > ### Author Response · Authors · 2022-11-18
> > > **Response to Reviewer uN6v**
> > >
> > > Dear Reviewer uN6v,
> > >
> > > We thank you for your valuable comments and suggestions to improve the manuscript. Should you have further questions, please let us know.
> > >
> > > Best,

---

### Official Review · Reviewer_qiuo · 2022-10-28

**Confidence:** 4
**Correctness:** 3
**Technical Novelty And Significance:** 1
**Empirical Novelty And Significance:** 2
**Recommendation:** 3

**Clarity, Quality, Novelty And Reproducibility:**

This paper is mostly clear and well written. Source code is provided for reproducibility. Please refer to my above comments for the major concern about lack of novelty.

**Strength And Weaknesses:**

Strength: this paper combines a set of existing (and indeed very popular) deep learning methods to deal with the numerical interpolation problem, and it achieves good results by experiments on real data sets.


Weakness: there are several concerns, among which lack of novelty is the major concern.

1. This paper combines a set of very popular techniques in deep learning to the numerical interpolation problem, such as transformer with self-attention and enhancement with pre-training. Every component of the proposed method is either existing or trivial, so it is not clear what is the novel method proposed by this paper.

2. The claimed contribution, the partial self-attention mechanism, is mostly trivial which sets zero correlation weights to target points (eq. (2)).  While the authors argued that such partial self-attention mechanism is connected to Radial Basis Function (RBF interpolation), it is really trivial to me: if one sets zero weights to target points, then it is naturally written as a combination of functions on the observed points. In addition, while observed and target points are decoupled, I believe it should be helpful to boost the performance by modeling the correlation among target points by common principles of semi-supervised learning (the target points are given).

3. Why are different metrics applied to different data sets, such as MSE for the NeSymReS, D30 and PhysioNet datasets, and MAE (CMAE,BMAE) for the TFRD-ADlet dataset?

**Summary Of The Paper:**

This paper proposes a learning-based approach using the encoder representations of Transformers for numerical interpolation of scattered data.

**Summary Of The Review:**

This paper combines existing techniques with trivial modification to achieve good results for numerical interpolation. Due to lack of novelty, I believe this is a good technical report for practitioners to apply deep learning methods for numerical interpolation, but by its current form it is not ready to be published a paper in a top machine learning venue.

---

> ### Author Response · Authors · 2022-11-11
> **Response to Reviewer qiuo (Part 1)**
>
> > **1.** This paper combines a set of very popular techniques in deep learning to ......
> >
> > **2.** The claimed contribution, the partial self-attention mechanism, is mostly trivial ......
>
> **Response:**
>
> We understand the reviewer's concerns on the novelty of our work. We agree with the reviewer in that Transformer and pre-training are indeed very popular techniques in CV and NLP. However, this does not mean that our work is trivial or not novel. The novelty of our work lies in the innovative application of Transformer encoder and pre-training for accurate scattered-data interpolation, which are detailed as follows.
>
> 1. **A novel framework suitable for interpolation tasks:**  The interpolation for scattered data has its own unique characteristic, i.e., both observed points and target points are indeed sampled from a common function, which leads to the inherent and significant correlation among them. This characteristic makes interpolation greatly different from the tasks in CV and NLP.
>
>      Effectively modeling and exploiting these correlations is a great challenge to accurate interpolation. The existing approaches process these two types of points separately and thus cannot effectively exploit the correlation. To address this problem, we propose a unified way to encode observed and target points and propose to use partial self-attention to avoid the interference of target points, which makes our interpolator more robust and generalizable. The experimental results show that our approach significantly outperforms all classical methods as well as previous data-driven approaches.
>
>     Scattered-data interpolation is a fundamental problem in scientific computing and engineering fields; thus, we believe that our work will have a large and widespread impact in these areas.
>
> 2. **The novel partial self-attention mechanism:** We propose a novel partial self-attention mechanism, which is extremely suitable for interpolation tasks. The partial self-attention sufficiently models and exploits the correlation between observed and target points in their common representation space, and avoids the interference caused by target points. In addition, it is experimentally effective and easy to implement.
>
>     More importantly, we interpret the partial self-attention mechanism as a feature interpolation layer using learnable basis functions by linking it to the traditional RBF interpolation approach. As shown in Equations 3 and 4, the connection that we present is clear and concise; but not very easy to discover in advance. Although considering partial self-attention as a linear combination is trivial, interpreting it as an interpolator using linear combinations of basis functions is not.
>
>     To the best of our knowledge, this study is the first work to propose partial self-attention and interpret it as an interpolating layer using learnable basis functions. Our work bridges the studies on interpolation algorithms and Transformers. This insight not only provides an explanation for our interpolation approach, but is also instructive for the interpretability of Transformers.
>
> 3. **The novel application of pre-training technique in scattered-data interpolation:** The use of pre-training technique has a significant contribution to overcome two limitations of the existing data-driven interpolation approaches, including:
>
>     a. The traditional interpolation approaches can be directly used without any training process; however, the existing data-driven interpolation methods require training on a sufficient dataset, making them difficult to deploy and use in a practical scenario.
>
>     b. In addition, the existing data-driven interpolation approaches were trained on a specific dataset, which limits their generalization to solve multiple scenario tasks.
>
>     To overcome these limitations, we propose to build pre-trained interpolation models using NIERT. Due to the lack of large-scale pre-training datasets for interpolation, we constructed the NeSymReS dataset, which contains a large variety of synthetic symbolic mathematical functions (see Supplementary Part B.1 for details). These functions approximate general and diverse function distribution, thus gaining our pre-trained model strong generalization ability. The experimental results shown in Tables 2 and 4 demonstrate the improvement of the interpolation accuracy of the pre-trained model for all datasets on practical applications.
>
>     Although pre-training techniques are commonly used in NLP and CV, this is a new idea for generalized scattered data interpolation. To the best of our knowledge, this study is the first work to propose the pre-trained models for scatter-data interpolation, we have verified that such interpolation pre-trained models can be generalized to a wide variety of interpolation tasks.
>
>     We realized that we have not clearly illustrated this point. We have expanded the description in the revised manuscript (Section 1 Page 2, and Section 3.3 Page 5).

---

> ### Author Response · Authors · 2022-11-11
> **Response to Reviewer qiuo (Part 2)**
>
> > **3.** In addition, while observed and target points are decoupled, I believe it should be helpful to boost the performance by modeling the correlation among target points by common principles of semi-supervised learning (the target points are given).
>
> **Response:**
>
> We understand the reviewer's idea that modeling the relationship among target points will improve the interpolation accuracy. However, this intuition is inappropriate for learning on scattered data interpolation tasks due to the following concerns:
>
> 1. **Modeling the correlation among target points is detrimental to the generalization of the interpolation model to the interpolation tasks with different numbers or distributions of target points.** Modeling the correlation among target points causes the model to overfit this correlation on the training set, which would make the model sensitive to the number or distribution of target points, and even prevent the model from generalizing to interpolation tasks with different numbers or distributions of target points.
>
> 2. **Modeling the correlation among target points has limited significance in improving the interpolation accuracy even on the test tasks with a consistent number and distribution of target points.** Although the observed and target points are decoupled in a certain interpolation task, they are resampled or re-divided randomly at each training session. This means that the model has seen enough observed points to represent the scattered data distribution over the long training period. At this point, modeling the correlation among target points is not more beneficial.
>
>
> To validate these ideas, we trained *NIERT using partial self-attention with the correlation among target points* and tested its interpolation accuracy under the following two different settings:
>
> 1. on the interpolation tasks in NeSymReS 1D test set with different numbers of target points. The number of target points of the test task ranges from $1$ to $2048$, while the number of target points during training ranges from $206$ to $246$.
>
> 2. on a modified NeSymReS 1D test set, where the target points is restricted in $[0,1]$. It should be noted that the target points are scattered in $[-1,1]$ in the tasks of the training set. This means that the target points in the test set are still in the domain, but their distribution changes.
>
>
> The experimental results using setting **1** show that when the correlation among target points is modeled, the accuracy of the model is very little improved on the test tasks of the same target points' number. Moreover, on test tasks with few target points (less than 64), there is a significant decrease in its accuracy. When the number of points increases (more than 1536), the interpolation accuracy also decreases to a certain extent. This is similar to the performance of the vanilla self-attention mechanism. We have updated this result in Figure 6 in Section 4.5.
>
> The experimental results using setting **2** are shown in the table below. These results show that the approach which models correlation among target points has a significant decrease in accuracy for interpolation tasks where the distribution of target points changes. In contrast, the accuracy of the model with partial self-attention is robustly held because interference among target points is avoided.
>
> | Attention type | MSE ($\times 10^{-5}$) on NeSymReS 1D testset | MSE ($\times 10^{-5}$) on *modified* NeSymReS 1D testset  |
> |-|-|-|
> | Partial self-attention | 8.964 | **9.001** |
> | Vallina self-attention  | 8.994 | 35.935 |
> | Partial self-attention  + target points' correlation | 8.959 | *28.827* |
>
> The results of these experiments indicate that modeling correlation among target points does not help to improve the interpolation accuracy, but makes it difficult for the model to generalize to the tasks with different target points' numbers or distributions. This is in line with our concerns.
>
>
> > **4.** Why are different metrics applied to different data sets, such as MSE for the NeSymReS, D30 and PhysioNet datasets, and MAE (CMAE,BMAE) for the TFRD-ADlet dataset?
>
> **Response:**
>
> MSE is the most commonly used metric for interpolation accuracy. Following the convention, we, therefore, used MSE for evaluation on NeSymReS, D30 and PhysioNet datasets.
>
> We use MAE for the TFRD-ADlet dataset to follow the convention of Ref. [1]. In this work, MAE is used as a metric to evaluate the accuracy of the model on the TFRD-ADlet dataset.
>
>
> **References**
>
> [1] Xiaoqian Chen, Zhiqiang Gong, Xiaoyu Zhao, Weien Zhou, and Wen Yao. A Machine Learning Modelling Benchmark for Temperature Field Reconstruction of Heat-Source Systems. arXiv preprint arXiv:2108.08298, 2021.

---

### Official Review · Reviewer_rgW7 · 2022-11-03

**Confidence:** 4
**Correctness:** 3
**Technical Novelty And Significance:** 3
**Empirical Novelty And Significance:** Not applicable
**Recommendation:** 6

**Clarity, Quality, Novelty And Reproducibility:**

The paper is well written, and the results are novel to the best of my knowledge. An anonymous link to the code is provided. There are a lot of experiments and ablation studies conducted both on synthetic and real-world datasets.

**Strength And Weaknesses:**

Strength:
A new mechanism to mask points in self-attention was introduced.
The proposed model achieves a better result on the synthetic and real-world dataset from the paper compared to the current state-of-the-art methods.
Extensive analysis and ablation studies were conducted.
The Paper is written in an easy to read manner
Strong evaluation and comparisons.

Weakness:
Mistypes in the text.
Some claims of the paper were not well-supported (see summary for more details).

**Summary Of The Paper:**

The paper introduces a new partial self-attention layer to improve numerical interpolation for scattered data. The module was used in a modified Transformer to solve the interpolation task. The proposed approach treats observed and target points in a unified way by embedding them in the same representation space.

**Summary Of The Review:**

In terms of text quality, the paper is easy to read. There are some minor issues with the text, but nothing major, except that the sentence abruptly ended at the end of page 4.

Experiments and evaluation design are done nicely; the authors conducted many experiments and ablation studies.

However, there are some issues with the paper:

The paper claims that one of the reasons why other methods perform poorly is because they process observed and target points separately. This fact prevents them from exploiting correlations between observed and target points (see page 2, end of paragraph 1, and page 5 before the experiments section). I do not see this claim proven in the paper, although I may be wrong.

There are a lot of variables (for example, different network parameter sizes) that can influence the outcome of this kind of experiment, so careful experiment design is needed to prove this.

In section 4.4 in figure 5, the fact that attention maps are imbalanced does not mean much. In my opinion, its area of affection for each point depends on its neighbors' count and their proximity placement. For example, if you have a lot of points in one area, it is natural that each point will contribute, or can either contribute a little to a wider area, or can contribute a lot to a local area. Which option is better is a good question. So the claim in the paper at the end of this section that NIERT can exploit the correlation between points observed and target points more effectively, in my opinion, is not well-supported.

Finally, the simple use of the pre-training technique is not a contribution.

Questions:
Is it correct that partly self-attention is just a particular case of self-attention, where observed and target points can’t “look” at the target points(wij=0)?
Please clarify, what is input in each batch for your neural network in evaluation time?


Small suggestions:
1. Section 3.4 is a good addition to the text but can be moved to supplementary.
2. It would be nice to see a comparison of several hyperparameters for each model for the methods you compare with.

---

> ### Author Response · Authors · 2022-11-11
> **Response to Reviewer rgW7 (Part 1)**
>
> We thank the reviewer for his/her approval of our work. The reviewer also raised several fundamental concerns which we addressed below.
>
> > **1.** sentence abruptly ended at the end of page 4.
>
> **Response:**
>
> We thank the reviewer for pointing out this error and we have corrected it.
>
> > **2.** The paper claims that one of the reasons why other methods perform poorly is because they process observed and target points separately. This fact prevents them from exploiting correlations between observed and target points (see page 2, end of paragraph 1, and page 5 before the experiments section). I do not see this claim proven in the paper, although I may be wrong.
>
> **Response:**
>
> We justify the claim as follows:
>
> 1. Methodology analysis:
>     The previous methods use an encoder-decoder architecture; they treat observed and target points separately, and they model and exploit the correlation in the decoder only. In contrast, our method NIERT treats observed and target points in a unified way and models and exploits the correlation in the same representation space at each layer.
>
> 2. Analysis of the contribution of observed points:
>     Figure 13 shows the contribution of observed points in the TFR-transformer, a representative of previous methods. As shown in this figure, only five observed points (say the points numbered 0, 2, 6, 16 and 19) have dominating contribution to target points in the whole domain while the other 17 observed points have little contribution, even for the target points in their close proximity. This fact clearly indicates that the correlation between observed points and target points cannot be fully modeled and exploited by TFR-transformer. We can achieve a similar observation from Figure 5 although it shows only four cases of representative observed points.
>
> 3. Experimental results:
>     The essential difference between NIERT and previous approaches is the way to process observed and target points. Therefore, the fact that NIERT outperforms all previous methods (especially the TFR-transformer) in terms of interpolation accuracy (Section 4.2) indicates that the previous methods didn't model and exploit the correlation of scattered data well.
>
> Together, these analyses and results demonstrate that the previous methods cannot model and exploit the correlation between observed and target points well.
>
> > **3.** There are a lot of variables (for example, different network parameter sizes) that can influence the outcome of this kind of experiment, so careful experiment design is needed to prove this.
>
> **Response:**
>
> We understand the reviewer's concern on the setting of hyperparameters. We have investigated the influence of various settings of these parameters (Tables 8 and 9) and chosen the optimal setting.
>
> It should be pointed out that, for the sake of fair comparison, we have ensured that our model and the TFR-transformer have the same setting of hyperparameters, including the number of attention layers, the dimension of the hidden space, the number of attention heads, etc. For other previous data-driven methods, whose architectures are quite different from ours, we used the parameter settings that were used by Ref. [1] and [2].
>
>
> > **4.** In section 4.4 in figure 5, the fact that attention maps are imbalanced does not mean much. In my opinion, its area of affection for each point depends on its neighbors' count and their proximity placement ......
>
> **Response:**
>
> We understand the reviewer's concern on the contribution of the observed points. Our claim is based on the following analysis and observations.
>
>
> 1. In principle, scattered data interpolation is expected to be local due to the continuity of the interpolation function. Hence, each observed point is expected to significantly affect its neighborhood, although the neighborhood might be large or small. Generally speaking, an observed point with a lot of neighbors has a small affection area while an observed point with few neighbors has a larger affection area.
>
> 2. Contribution analysis of observed points shows that, for TFR-transformer, only five observed points (say the points numbered 0, 2, 6, 16 and 19) have dominating affection to the whole domain (Figure 13), while the other 17 observed points have little affection, even for the target points within their proximity. These unbalanced responses mean that TFR-transformer deviates from the local nature of scattered data interpolation.
>
> 3. On the contrary, in NIERT, each observed point has significant affection to a large or small area in its proximity (Figure 13), which is consistent with the local nature of scattered data interpolation.
>
> These analyses and observations well support the claim that NIERT can exploit the correlation between observed and target points more effectively.

---

> ### Author Response · Authors · 2022-11-11
> **Response to Reviewer rgW7 (Part 2)**
>
> > **5.** Finally, the simple use of the pre-training technique is not a contribution.
>
> **Response:**
>
> We understand the reviewer's concern on the contribution of the pre-training technique used by NIERT. We realized that we didn't clearly illustrated this point. We have expanded the description in the revised manuscript (Section 1 Page 2, and Section 3.3 Page 5) and explain it in detail as follows.
>
> The use of pre-training technique has a significant contribution to overcome two limitations of the existing data-driven interpolation approaches, including:
> 1. The classical interpolation methods can be directly used without any training process; however, the existing data-driven interpolation methods require training on sufficient data, making them difficult to deploy and use in a practical scenario.
>
> 2. In addition, the existing data-driven interpolation approaches were trained on a specific dataset, which limits their generalization to solve multiple scenario tasks.
>
> To overcome these limitations, we propose to build pre-trained interpolation models using NIERT. Due to the lack of large-scale pre-training datasets for interpolation, we constructed the NeSymReS dataset, which contains a large variety of synthetic symbolic mathematical functions (see Supplementary Part B.1 for details). These functions approximate general and diverse function distribution, thus gaining our pre-trained model strong generalization ability. The experimental results in Table 2 and 4 demonstrate the improvement of the interpolation accuracy of the pre-trained model for all datasets on practical applications.
>
> Although pre-training techniques are commonly used in NLP and CV, this is a new idea for generalized scattered data interpolation. To the best of our knowledge, this study is the first work to propose the pre-trained models for scatter-data interpolation, we have verified that such interpolation pre-trained models can be generalized to a wide variety of interpolation tasks.
>
> > **6.** Questions: Is it correct that partly self-attention is just a particular case of self-attention, where ......
>
> **Response:**
>
> Yes, we completely agree with the reviewer in that partial self-attention is "a particular case of self-attention, where observed and target points can’t look at the target points".
>
> It is worth pointing out that partial self-attention is extremely suitable for interpolation tasks. In contrast, vanilla self-attention suffers from interference from target points to observed points. The ablation experiments (Section 4.5 Figure 6) clearly illustrate that vanilla self-attention leads to poor generalization ability.
>
> > **7.** Please clarify, what is input in each batch for your neural network in evaluation time?
>
> **Response:**
>
> In evaluation time, each batch consists of multiple interpolation tasks, and each interpolation task is a 2-tuple $(O,T)$, where $O= \\{(x_i, y_i)\\}\_1^{n}$  represents the set of $n$ observed points and $T=\\{x_i\\}_{n+1}^{n+m}$ represents the set of $m$ target points.
>
> > **8.** Small suggestions: 1. Section 3.4 is a good addition to the text but can be moved to supplementary.
>
> **Response:**
>
> We thank the reviewer for his/her pointing out the good addition of Section 3.4 to the text. Section 3.4 aims to illustrate the deep connection between NIERT and classical interpolation methods. It shows that partial self-attention is a general and learnable form of RBF interpolation, which provides a plausible explanation of our approach. This is why we opted to put Section 3.4 into the main text.
>
> > **9.** Small suggestions: 2. It would be nice to see a comparison of several ......
>
> **Response:**
>
> We appreciate the reviewer's suggestion to "compare several hyperparameters for each model for the methods we compared with". These methods differ greatly in their architecture; thus, we have divided these methods into three categories for comparison:
>
> 1. For the TFR-transformer, we set its hyperparameters identical to NIERT. These  hyperparameters include hidden space dimension, the number of attention layers and the number of attention heads, which are listed in Supplementary Table 5.
>
> 2. For CNP, ANP, and BANP, their architectures differ greatly from NIERT; thus, we set their hidden space dimension to be the same as NIERT only. For a fair comparison, we set the other parameters of these methods identical to Ref. [1].
>
> 3. For RNN-VAE, L-ODE-RNN, L-ODE-ODE, and mTAND-Full, we use the same hyperparameter settings as Ref. [2].
>
> We have now inserted new sentences to clearly illustrate the hyperparameter settings of these methods (Part C of Supplementary Text).
>
> **References**
>
> [1] Xiaoqian Chen, Zhiqiang Gong, Xiaoyu Zhao, Weien Zhou, and Wen Yao. A Machine Learning Modelling Benchmark for Temperature Field Reconstruction of Heat-Source Systems. arXiv preprint arXiv:2108.08298, 2021.
>
> [2] Satya Narayan Shukla and Benjamin Marlin. Multi-Time Attention Networks for Irregularly Sampled Time Series. ICLR, 2021.

---

### Author Response · Authors · 2022-11-18
**General comments and change list**

Dear reviewers,

We thank all of you for your valuable comments and suggestions to improve our manuscript. Following the suggestions, we have revised the manuscript. All changes have been highlighted in red.

Here we summarize the changes as follows:
- Adding an ablation experiment for the partial self-attention mechanism
  - Following the suggestion of Reviewer qiuo, we have carried out an experiment to assess the impact of considering correlation among target points in partial self-attention (see Section 4.5 and Figure 6).
- Further elucidating the novelty of the use of the pre-training technique
  - Following the comments of Reviewers rgW7 and qiuo, we have further elucidated the novelty of the use of the pre-training technique (see Section 1 and Section 3.3 for details).
- Adding a table to show statistical information of the four datasets
  - As suggested by Reviewer uN6v, we have further tabulated the statistical information on the number of scattered points for interpolation tasks in each dataset (see Supplementary Table 6).
- Adding descriptions of hyperparameter settings for each approach used for comparison
  - Following the reviewer rgW7's suggestion, we have inserted sentences to illustrate the hyperparameter settings of the approaches used for comparison (see Supplementary Part C).
- Adding discussion of computational cost and accuracy trade-off
  - Following the suggestion of Reviewer uN6v, we have inserted a new section to discuss the time spent on training and interpolation of different approaches. We also visualized the trade-off between accuracy and interpolation time. (see Supplementary Section D.5).
- Improving the contribution analysis of observed points
  - Following the comments of Reviewer rgW7, we have improved the contribution analysis of observed points (see Supplementary Section D.2).

We hope that our answers and revision of the paper help clear up your doubts.

Best,

The authors

---

### Decision · Program_Chairs · 2023-01-20

**Decision:**

Reject

**Justification For Why Not Higher Score:**

N/A

**Justification For Why Not Lower Score:**

N/A

**Metareview: Summary, Strengths And Weaknesses:**

We would like to thank the authors for submitting the paper and the responses and revision during the rebuttal period.
Unfortunately, the recommendation for this paper is a Reject.
The reviewers feel that the paper, although introducing some interesting concepts such as new attention masking and utilizing transformers for interpolation, still does not present the level of contribution expected from an ICLR paper. We encourage the authors to further develop their method and theoretical analysis and resubmit to a future conference.